# New Alternatives in the Valorisation of *Eucalyptus globulus* By-Products for the Textile Industry

**DOI:** 10.3390/polym17121619

**Published:** 2025-06-11

**Authors:** Jorge Santos, Carlos Silva, Raquel A. Fernandes, Nuno Ferreira, Danilo Escobar-Avello, Pedro Magalhães, Fernão D. Magalhães, Jorge M. Martins, Luisa H. Carvalho

**Affiliations:** 1ARCP-Associação Rede de Competência em Polímeros, 4200-355 Porto, Portugal; raquel.fernandes@arcp.pt (R.A.F.); nuno.ferreira@arcp.pt (N.F.); 2LEPABE—Faculty of Engineering, University of Porto, Rua Dr. Roberto Frias, s/n, 4200-465 Porto, Portugal; fdmagalh@fe.up.pt (F.D.M.); jmmartins@estgv.ipv.pt (J.M.M.); lhcarvalho@estgv.ipv.pt (L.H.C.); 3ALiCE—Associate Laboratory in Chemical Engineering, Faculty of Engineering, University of Porto, Rua Dr. Roberto Frias, 4200-465 Porto, Portugal; 4Tintex Textiles SA, Zona Industrial, Polo 1, Campos, 4924-909 Vila Nova de Cerveira, Portugal; carlos.silva@tintextextiles.com (C.S.); pedro.magalhaes@tintextextiles.com (P.M.); 5Unidad de Desarrollo Tecnológico, Universidad de Concepción, Coronel 4191996, Chile; daniescobar01@gmail.com; 6Centro Nacional de Excelencia para la Industria de la Madera (CENAMAD), Pontificia Universidad Católica de Chile, Av. Vicuña Mackena 4860, Santiago de Chile 8331150, Chile; 7DEMad-Department of Wood Engineering, Instituto Politécnico de Viseu, Campus Politécnico de Repeses, 3504-510 Viseu, Portugal

**Keywords:** functional textiles, vegan textiles, coated fabrics, biomass, antioxidant, value chain, low impact development, sustainable materials

## Abstract

The textile industry requires products with a wide range of characteristics for use in diverse applications such as the production of shoes, bags, jackets, thermal clothing and articles for the automotive industry, among others. These products have traditionally been made from leather, which is obtained from animal hides. However, leather production has come under enormous pressure due to sustainability concerns in various areas and the growing number of people who actively choose to avoid all animal products. The main solutions developed by the textile industry have been to apply synthetic coatings based on polyvinyl chloride (PVC) or polyurethane (PUR) to textile substrates. One of the ways to reduce the environmental impact and non-renewable content of artificial leather is to replace parts of the synthetic component with lignocellulosic by-products. In the present work the feasibility of using small branches and leaves of *Eucalyptus globulus* (BLE) as a component of an aqueous PUR formulation for coating textile products was evaluated. In addition, the possibility of obtaining functional textile products with antioxidant properties based on the BLE particles incorporation was also evaluated. The effect of the BLE particle size distribution in the PUR formulation and on the properties of the coated textile products was evaluated. The BLE particles and their size influenced the colour, appearance, hydrophobicity and mechanical properties of the coated textiles. The (BLE) particles have improved the tensile strength of textile coating products without loss of elongation, improving their properties for specific applications. Furthermore, the textiles coated with the (BLE) particles showed interesting antioxidant properties, being possible to obtain coated fabrics with five times more DPPH radical scavenging activity than the reference coated fabric without (BLE) particles.

## 1. Introduction

The textile industry has recently shown an increased interest in developing new products using forest raw materials, driven by concerns for the environment. The main advances have been in the use of eucalyptus wood to produce a semi-synthetic fibre called Lyocell. However, the forestry industry produces a large quantity of by-products, mainly bark, branches and leaves, with the valorisation of eucalyptus bark studied previously with interesting applications in substrates for the germination of plants [1,2], in wood-based panels [3,4], or for insulation solutions [5]. However, small branches and leaves (BLE) do not have the same interest and are left in the forest without further valorisation.

As reported in previous studies, it is possible to use (BLE) particles as a raw material to obtain extracts that allow dyeing and functionalising cotton fabrics [6].

This study focuses on the global valorisation of (BLE) particles by using them as a raw material in the production of new functional textiles.

The textile industry is always demanding new products with very differential characteristics to be used in conditions as diverse as in the production of shoes, bags, jackets, thermal clothing, items for the automobile industry, among others, and historically, these kinds of products were made from leather obtained from the skins of animals. However, the production of leather has been under enormous strain due to worries about sustainability in various fields (due to greenhouse gas emissions from livestock farming) and the rising number of people who choose to actively eat meatless or abstain from all animal products [7]. As a result, alternative vegan alternatives have become necessary.

The main solutions developed by the textile industry were the application of synthetic coatings based on polyvinyl chloride (PVC) or polyurethane (PUR) on textile substrates. One of the possibilities to reduce the environmental impact and non-renewable content of artificial leather is to replace parts of the synthetic component with products derived from agricultural lignocellulosic by-products. In this way, interesting commercial solutions were developed using materials such as grape (Vegea SRL), apple, lemon, barley skin (Vegatex Biotech (HK) LTD.), or grape pomace (Tintex Textiles) [7]. However, few published studies have focused on evaluating the impact of industrial waste or by-products on textile coating formulations and the coated textile products produced [8,9,10].

In terms of research focusing on the incorporation of forestry by-products into textile coatings, recent work has demonstrated the potential of radiata pine sawdust as a component of textile coatings [11]. However, to the best of our knowledge, no study has been published evaluating the impact of BLE particles as a component of coating textile formulations.

The incorporation of a lignocellulosic material into a polymeric formulation, such as the PUR textile coating formulation, will mainly affect the rheological properties of the formulation and its reactivity. In addition, the variability of lignocellulosic by-products in chemical composition and morphology, due to their natural origin, necessitates a deeper understanding of the effect of the particles on the properties of the coating formulation, on the production process and on the properties of the final coated textile products, in order to ensure the reproducibility required for industrial application.

Aqueous dispersions of waterborne polyurethanes (WPUs) are widely used in textile applications and were the solution adopted in the study. Waterborne polyurethanes are less toxic and more environmentally friendly than their solvent-based counterparts, as they contain no hazardous volatile organic compounds [12]. For this reason, companies with strong environmental commitments choose to use these WPUs as the main components for textile coating formulations.

Lignocellulosic by-products are mainly composed of cellulose, lignin, and an extractable fraction, the nature and composition of which will depend on the nature of the by-product and the extraction conditions used [4].

Incorporation of the lignocellulosic by-product into the water-based textile coating formulation is achieved by mixing it with the WPU solution under continuous stirring. The extractable compounds present in the lignocellulosic material are “extracted” and migrate into the water-based WPU solution, changing the colour, the rheology, reacting with the polymeric components or imparting special properties to the coated textile product, such as antioxidant or antifungal properties.

Recent work carried out by the research group has shown that (BLE) particles contain approximately 19–25% water-extractable components, primarily hydrolysable tannins and sugars, and, to a lesser extent, condensed tannins [6]. The antioxidant properties of hydrolysable and condensed tannins are well known [4,13] and the possibility of transferring them to the textile product has been demonstrated by using (BLE) extracts to dye cotton fabrics [6]. In this case, it was expected that functional antioxidant textile products could be obtained by incorporating (BLE) particles into the coating textile formulation. Functional textile products with antioxidant properties have applications in the production of medical, food packaging, and sports textiles [14,15].

In this study, the BLE particles were air-dried, milled, and sieved to obtain five fractions with different particle size distributions. The morphology of the (BLE) particles and their thermal behaviour were characterised by SEM-EDS and TGA techniques. FTIR spectroscopy was used to analyse the effect of particle size on the chemical nature of the (BLE) samples. In addition, the stability of the (BLE) particles was tested by analysing the phenolic, sugar, and protein content of the (BLE) particles and their evolution with storage time. The cotton fabrics were coated with a polymer formulation based on the (BLE) particles and an aqueous polyurethane solution.

Therefore, this study has focused on the production of coatings for the textile industry incorporating (BLE) particles. The aim is to valorise the (BLE) particles, reduce the synthetic polymers used, and obtain coated textile products with different properties in terms of appearance, colour, and feel. In addition, the (BLE) was expected to impart interesting antioxidant properties to the final product [16].

## 2. Materials and Methods

### 2.1. Raw Material

*E. globulus* (BLE) branches and leaves were collected from a forest plantation in Ponteareas, NW Spain (42°11′22.5” N, 8°31′55.6” W), then dried until moisture content reached 10.3 ± 0.9% (105 °C), and finally ground in a cutting mill (Retsch, Haan, Germany) using a 1 mm sieve. Subsequently, the (BLE) particles were sieved in a vibratory sieve shaker (Retsch, Haan, Germany) yielding 6 samples with different particle size distributions: BLE32 (<32 µm), BLE32 (<32 µm), BLE63 (32–63 µm), BLE125 (63–125 µm), BLE200 (125–200 µm), and BLE500 (200–500 µm).

Samples of different sizes were made to observe the effect this has on the physical and mechanical properties of the coated fabric. An interlock (2740/3) 100% cotton at (447 gsm) from Tintex Textiles^®^ (Vila Nova de Cerveira, Portugal) was used as the textile substrate.

### 2.2. Fourier Transform Infrared Spectroscopy (FTIR) Assay

FTIR spectra were recorded on a VERTEX 70 FTIR spectrometer (Bruker, Billerica, MA, USA), which was equipped with a high-sensitivity DLaTGS detector (Bruker, Billerica, MA, USA), at room temperature.

Samples were measured in ATR mode using an A225/Q PLATINUM ATR diamond crystal (Bruker, Billerica, MA, USA) with a single reflection accessory. Spectra were recorded from 4000 to 400 cm^−1^. The resolution was 4 cm^−1^. All spectra were recorded and processed using OPUS 7.0 software.

### 2.3. Scanning Electron Microscopy

The surface morphology of the (BLE) particles and of the cotton fabrics coated with them was using SEM analysis. Cross-sections and surfaces of the coated cotton knits were scanned at different places on the sample using a Phenom XL scanning electron microscope (SEM, Thermo Fisher Scientific, Waltham, MA, USA) at magnifications of 250× and 2000×. Prior to analysis, the samples were coated with a gold–palladium (Au–Pd) layer to ensure conductivity. Energy dispersive spectroscopy (EDS) was then used to analyse the chemical composition of the (BLE) particles and the coated cotton fabrics.

### 2.4. Chemical Characterisation of (BLE) Particles

#### 2.4.1. Phenol Content

The Folin–Ciocalteu method is based on the formation of a blue-coloured complex resulting from the reaction between phenolic compounds and the Folin–Ciocalteu reagent in an alkaline medium. In this assay, 0.25 mL of sample was mixed with 15 mL of distilled water and 1.25 mL of diluted Folin–Ciocalteu reagent (1:10 *v*/*v*). After 8 min of reaction in the dark, 3.75 mL of sodium carbonate solution (75 g/L) was added, and the volume was brought to 25 mL with distilled water. The mixture was homogenized and incubated at room temperature, protected from light, for 2 h. Absorbance was then measured at 760 nm using a P9 UV/VIS double-beam spectrophotometer (Avantor VWR (Shanghai) Co., Ltd., Shanghai, China). A calibration curve was constructed using gallic acid standards in the range of 0 to 600 mg/L, and the total phenolic content was determined from this curve. Results were expressed as grams of gallic acid equivalents per 100 g of dry extract (g GAE/100 g dry extract) [17].

#### 2.4.2. Sugar Content

The quantification of reducing sugars was carried out using the anthrone method, which is based on the formation of a green-blue chromogenic complex resulting from the reaction between anthrone and carbohydrates under strongly acidic and high-temperature conditions. In this method, 2 mL of sample was added to 4 mL of cold sulfuric acid solution (75%). Then, 8 mL of freshly prepared anthrone solution was added. The tubes were sealed, vortexed, and heated in a boiling water bath (100 °C) for 15 min. After cooling to room temperature, absorbance was measured at 620 nm using a P9 UV/VIS double-beam spectrophotometer. A calibration curve was constructed using glucose standard solutions ranging from 10 to 100 mg/L, and the concentration of reducing sugars in the samples was determined from the curve. Results were expressed as grams of glucose equivalents per 100 g of dry extract (g GE/100 g dry extract).

#### 2.4.3. Protein Content

The Lowry method is based on the reaction between peptide bonds and copper ions under alkaline conditions, followed by the reduction of the Folin–Ciocalteu reagent by aromatic amino acids such as tyrosine and tryptophan, forming a blue-coloured complex. To perform the assay, 1 mL of NaOH solution (2 N) was added to 1 mL of the sample, and the mixture was hydrolysed at 100 °C for 10 min. After cooling to room temperature, 10 mL of freshly prepared Lowry reagent was added, and the solution was left to stand for 10 min in the dark. Then, 1 mL of Folin–Ciocalteu reagent (1 N) was added, followed by vortexing. The reaction mixture was incubated at room temperature for 30–60 min, protected from light. Absorbance was measured at 750 nm using a P9 UV/VIS double-beam spectrophotometer (Avantor VWR (Shanghai) Co., Ltd., Shanghai, China). The calibration curve was established using bovine serum albumin (BSA) standards in the concentration range of 0 to 1000 ppm. Results were expressed as grams of BSA equivalents per 100 g of dry extract (g BSA/100 g dry extract).

#### 2.4.4. Extraction Yield

Extraction experiments were conducted using an ultrasonic bath with temperature control. (BLE) particles were combined with water at a fixed solid-to-liquid ratio of 1:10 (*w*/*w*) in a 500 mL bottle. The bottle was immersed in the ultrasonic bath, which was pre-set to 35 kHz and 80 °C. After 30 min of sonication, the suspension was filtered to separate the solid from the liquid. The solid was rinsed with water until the filtrate was nearly colourless. The liquid was used to evaluate the phenolic, sugar, and protein content, while the solid was used to evaluate the extraction yield.

The extractions were performed in triplicate, and the analysis results (phenolic content, sugar content, protein content, and extraction yield) were averaged.

The extraction yield was determined by drying the solid fraction obtained post-extraction in a laboratory oven at a temperature of (60.0 ± 1.0) °C for 24 h. This procedure was conducted in quintuplicate, and the results were averaged.

The extraction yield was calculated by subtracting the weight of the dry material at the conclusion of the process from its weight at the beginning (Equation (1)).(1)$$EY\left(\%\right)=\frac{Dry\;material\;weight\;at\;the\;start(g)-Dry\;material\;weight\;at\;the\;endl(g)}{Dry\;material\;weight\;at\;the\;start(g)}\times 100$$

### 2.5. Coating of Cotton Fabrics with (BLE) Particles

For the coating formulation, an industrial textile coating formulation was used as a standard (PB) and the effect of adding the (BLE) samples (of different particle size distributions) was evaluated by replacing a percentage of the formulation components with the (BLE) particles (Table 1).

The coating formulations were prepared in an open beaker of 1000 mL, stirred by a mechanical stirrer (IKA Eurostar 20(IKA-Werke GmbH & Co. KG, Staufen, Germany) equipped with a 4-bladed propeller stirrer. The different components of the formulation, i.e., aqueous solution of polyurethane (WPU), blocked isocyanate and rheological additives, and BSG particles (when necessary), were added one by one, waiting to obtain a homogeneous mixture in each step.

Subsequently, the coating formulation was foamed for 1.5 min to obtain a stable foam.

The coating formulation with and without the BLE particles was applied on the textile substrate using a scraper to ensure uniformity in the coating. Subsequently, the coated textile was dried at 100 °C for 4 m s in a LABDRYER type «LTE» (WERNER MATHIS AG, Oberhasli, Switzerland). The dried coated textile was pressed in a single application compact calender H02 (MONTI ANTONIO S.P.A. Thiene, Italy), and the polymerisation process was performed in the same lab-dryer equipment at 150 °C.

### 2.6. Coated Fabrics Characterisation

#### 2.6.1. Colour and Colour Fastness Evaluation

The colour coordinates were assessed using a Ci7600 Sphere benchtop spectrophotometer (X-rite, Grand Rapids, Michigan) in accordance with the CIELab colour coordinate system. Measurements for the coordinates L* (representing lightness on the black/white axis), a* (indicating the red/green axis), and b* (reflecting the yellow/blue axis) were taken five times for each sample. The average values of these measurements were utilized for analysis.

The values of the total colour difference (ΔE), and the variation of the colour coordinates ΔL*, Δa*, and Δb*, were calculated to evaluate the impact of the (BLE) samples used in the textile coating formulation (ΔE). The total colour differences were determined using the equation provided in Equation (2).
(2)$${\Delta E}_{01}=\sqrt{{\left(L_{0}^{*}-L_{1}^{*}\right)}^{2}+{\left(a_{0}^{*}-a_{1}^{*}\right)}^{2}+{\left(b_{0}^{*}-b_{1}^{*}\right)}^{2}}$$
where L_0_*, a_0_*, and b_0_* represent the colour coordinate values of the PB textile coating produced using only the WPU formulation. The L_1_*, a_1_*, and b_1_* represent the colour coordinates of the coated textile products produced incorporating the (BLE) particles.

The light fastness of the coated fabrics was assessed by exposing the samples to a 6500 K lamp with an output of 2000 lumens for one week. The samples were subsequently measured to determine any alterations. To quantify the colour fastness, the total colour difference (ΔE) and the changes in colour coordinates (ΔL*, Δa*, and Δb*) were calculated using a designated equation (Equation (3)):
(3)$${\Delta E}_{T}=\sqrt{{\left(L_{BLExT}^{*}-L_{BLEx}^{*}\right)}^{2}+{\left(a_{BLExT}^{*}-a_{BLEx}^{*}\right)}^{2}+{\left(b_{BLExT}^{*}-b_{BLEx}^{*}\right)}^{2}}$$
where L_BLEx_*, a_BLEx_*, and b_BLEx_* are the colour coordinate values of the original coated cotton knit untreated, and L_BLExT_*, a_BLExT_*, and b_BLExT_* are the colour coordinates of the same coated cotton knit after the light treatment.

#### 2.6.2. Surface Wettability

Contact angles of water and hexane on the materials’ surfaces were measured. Contact angles were measured using a Dataphysics OCA 20 Plus (DataPhysics Instruments, Filderstadt, Germany) equipment, equipped with a single direct dosing system SDDM, and the data were analysed with the Dataphysics software SCA 20. A 4 μL drop was added to the coated textile surface and the contact angle after 120 s was measured. For each solution, five measurements were performed, and the results were averaged.

#### 2.6.3. Differential Scanning Calorimetry (DSC)

DSC analysis was performed on a NETZSCH DSC instrument (214 Polyma, Selb, Germany). Samples (7 ± 3 mg) were sealed in non-hermetic aluminium pans, crimped with an inverted cover. The temperature range was scanned from 25 °C to 500 °C at a heating rate of 10 °C min^−1^. Temperature and enthalpy calibrations were performed using indium calibration standards (purity > 99.999%).

#### 2.6.4. Simultaneous Thermal Analysis (STA)

TGA and DSC analyses were performed using a NETZSCH STA 449 F3 Jupiter^®^ (214 Polyma, Selb, Germany), (which allows measurement of mass changes (Thermogravimetric Analysis (TGA)) and thermal effects (Differential Scanning Calorimetry (DSC)). The technique was used to evaluate the thermal degradation of (BLE) particles using an atmosphere of N_2_ at a heating rate of 10 K/min from 50 to 800 °C, and was also used to characterise the thermal behaviour of coated and uncoated textile samples by heating the samples at a rate of 10 K/min under airflow between 25 °C and 500 °C.

#### 2.6.5. Evaluation of Antioxidant Capacity of the Fabrics

##### DPPH (2,2-diphenyl-1-picrylhydrazyl) Assay

A 10 mL solution containing 0.1 mM DPPH was added to 50 mg of each textile sample (both uncoated and coated with either the WPU formulation alone or the BLE particles). The samples were then incubated at room temperature in the dark for 1, 2, 4, or 6 h to evaluate their antioxidant activity. The solutions were measured at predefined time intervals at a wavelength of 515 nm using a VWR^®^ P9 UV/VIS double-beam spectrophotometer (Avantor VWR (Shanghai) Co., Ltd., Shanghai, China). Equation (4) was then used to determine the ‘DPPH radical scavenging activity (%)’
(4)$$‘DPPH\;radical\;scavenging\;activity\;(\%)’=\frac{{Absorbance}_{control}-{Absorbance}_{sample}}{{Absorbance}_{control}}\times 100$$
where the control was the DPPH solution incubate without sample.

#### 2.6.6. FRAP Essay

To evaluate the antioxidant activity of the uncoated fabrics and those coated with the WPU formulation alone (PB) and with the (BLE) particles (BLE32-200) using the ferric reducing antioxidant power (FRAP) method, samples of (1.0 ± 0.1) g of the fabric were cut and mixed with 25.0 mL of NaOH (1 M) solution in a sealed bottle (50 mL). The bottles were then placed in an ultrasonic bath (Sonorex Super RK 512 H, Bandelin, Berlin, Germany) (frequency 35 kHz), which had previously been pre-set at 80 °C. After 60 min of contact, the suspension was vacuum filtered, the fabric was washed with water and dried in an oven (60 ± 5 °C) for 24 h. The liquid fraction was used to evaluate the antioxidant activity using the FRAP method.

The FRAP assay evaluates the ability of antioxidants present in a sample to reduce yellow ferric ions (Fe^3+^) to blue ferrous ions (Fe^2+^) under acidic conditions. To perform the assay, 12.0 mL of freshly prepared FRAP reagent (composed of acetate buffer, TPTZ solution, and FeCl_3_ solution) was mixed with 0.40 mL of the sample solution. After 5 min of incubation at room temperature in the dark, absorbance was measured at 593 nm using a VWR^®^ P9 UV/VIS Double-Beam Spectrophotometer (Avantor VWR (Shanghai) Co., Ltd., Shanghai, China). A calibration curve was constructed using ascorbic acid standards in the concentration range of 0.1 to 1.0 mmol/L, and the antioxidant capacity of the samples was calculated based on this curve. Results were expressed as milligrams of ascorbic acid equivalents per 100 g of dry extract (mg AAE/100 g dry extract).

### 2.7. Statistical Analysis

All measurements were conducted in triplicate and the results are expressed as means ± standard deviations. The influence of (BLE) particle size on the properties of the samples (protein content, phenols, sugars, among others) and on the properties of the textile products coated with these particles was statistically analysed by a one-way analysis of variance (ANOVA) with a significance level set at *p* < 0.05. The Real Statistics Resource Pack and Tukey’s test were used to assess differences between means. Real Statistics Resource Pack and Tukey’s test were used to assess differences between means.

## 3. Results and Discussion

### 3.1. Characterisation of E. globulus Branches and Leaves

Milling is one of the first steps in the processing of lignocellulosic by-products and will influence their applicability in terms of particle size distribution and morphology. In this case, a knife mill with a 1 mm sieve was selected for the study.

The particle size distribution of the dry and milled particles (BLE) was evaluated by gravimetrically fractionating the particles in six samples by mechanical sieving.

The percentage in weight of each sample is shown in Figure 1.

Despite the use of a 1 mm sieve, the majority fraction was between 200 and 500 μm, followed by fractions between 63 and 200 µm, while particles smaller than 63 µm and larger than 500 µm had the lowest percentage of particles. The particle size fractions of <32 µm (BLE32), of 32–63 µm (BLE63), 63–125 µm (BLE125), 125–200 µm (BLE200), and 200–500 µm (BLE500) were used to the study of the impact of (BLE) particles on the textile coating formulation.
*Thermal characterisation of (BLE) particles by TG*

The thermal degradation of (BLE) unsieved particles was evaluated employing TGA analysis in N_2_ atmosphere with a heating rate of 10 K/min from 50 to 800 °C (Figure 2).

The (BLE) particles were observed to undergo five main stages of degradation.

The first peak of weight loss was observed at 85.4 °C, related to the evaporation of moisture and low temperature volatile compounds. The second and third peaks of degradation were observed at 241.6 °C, related to the decomposition of hemicellulose, and at 338.4 °C, due to the degradation of cellulose [18]. Another peak of weight loss was observed at 408.5 °C, associated with the degradation of low molecular weight polyphenols present in the (BLE) particles. Finally, a slow and continuous degradation was observed between 500 °C and 800 °C, attributed to the degradation of lignin. Residue mass was 24.2% at 800 °C and approximately 27.0% at 600 °C, a similar value as previously observed by other authors (26.0%) when evaluating commercial bleached eucalyptus kraft pulp fibres [18].
*Chemical characterisation of (BLE) particles and storage time impact*

One of the main barriers to the viability of industrial application of a process based on lignocellulosic products is ensuring material stability and reproducibility of the newly developed industrial process. To this end, the effect of storage time on the properties of unsieved (BLE) particles was investigated.

The milled (BLE) particles were stored in black plastic bags (to avoid light exposure) in a climate-controlled room at (20 ± 5) °C for 12 months.

To assess the effect of time on the chemical composition of the (BLE) particles, four chemical characterisations were performed, the first immediately after milling (M0), the second after 4 months of storage (M4), the third after 8 months of storage (M8) and the last before 12 months of storage (M12).

To evaluate the chemical composition of the (BLE) particles, an ultrasonic water extraction was performed at 80 °C for 30 min, according to a previously published method [19].

The extraction yield, phenolic, sugar content, and protein content were evaluated, and the results are shown in Figure 3.

(BLE) particle extracts contained 246.1–278.4 mg GAE/g extract (phenols), 184.6–244.0 mg of GE/g of extract (sugars) and 228.8–391.1 mg BSA/g extract (proteins), with the phenolic content similar to that obtained for ethanolic extracts of *E. globulus* leaves (273.2 ± 17.5 mg GAE/g extract) by other authors [19] and the protein content higher than that previously reported by other authors for *E. globulus* leaves (218 mg ABSE/g extract) and seeds (151 mg ABSE/g extract) [20].

Regarding the effect of storage time, phenolic and sugar contents were relatively stable, with a significant decrease in sugar content only after twelve months of storage (Figure 3). However, the protein content was the component most sensitive to storage, as it registered a significant decrease over time (Figure 3).

The extraction yield of (BLE) particles after 4 months of storage was significantly improved, when compared to the initial value (M0, Figure 3) due to the degradation of high molecular weight water insoluble compounds and remained stable up to 8 months of storage (Figure 3). After 12 months, a decrease was observed (Figure 3), possibly due to the loss of small phenols/organics.

#### 3.1.1. Characterisation of the (BLE) Particles by SEM Microscopy

The particle morphology influences its interaction with the polyurethane (WPU) matrix and its stacking properties during the mixing, application, drying, and pressing stages of the textile coating formulation. SEM images of (BLE) samples with different particle sizes are shown in Figure 4.

Regarding the analysis of particle morphology, it was possible to identify two main types of particles, one more fibrous and the other with an amorphous nature (Figure 4).

As for the non-fibrous particles, the stomatal pores of the eucalyptus leaves could be distinguished in the higher granulometry samples (BLE200 and BLE500) [21], but not in the lower granulometry samples, which is due to the higher degree of grinding.

The particle size of the SEM images obtained at 250× magnification, and the results of average particle size are shown in Table 2.

The amorphous particles have a particle size between the sieves used, but the fibre particles were found to be longer (Table 2). This fact may be associated to the mechanical degradation caused by the fibres passing through the sieves.

The (BLE) particles are a mixture of fibrous and amorphous particles with the estimated number of fibres being 25.0% for BLE500, 27.3% for BLE200, 10.5% for BLE125, 8.3% for BLE63, and less than 1% for BLE32.

In addition, the chemical composition of the (BLE) sample particles was determined by EDS, and the results are shown in Table 3.

The leaves and branches of *E. globulus* have different carbon, nitrogen and oxygen contents. Leaves have a higher carbon content than branches, with an average of 52.9%. Branches and wood have a lower content, around 46.8% [22]. Nitrogen content varies from leaves to branches and also depends on age and month of harvesting, with the N content of *Eucalyptus Globulus* leaves being around 7–33 (mg seedling^−1^) and for stems around 3–13 (mg seedling^−1^) [23]

The physical and chemical interaction between the particles and the WPU coating formulation depends on the morphology of the particles and their chemical nature. As mentioned previously, (BLE) particles contain soluble proteins, polyphenols, sugars, and waxes that can react with the reactive isocyanate and hydroxyl groups present in the WPU formulation.

The proportion of N and O groups present in the (BLE) particles will affect their chemical interactions with the isocyanate and hydroxyl-free groups present in the polyurethane coating formulation. In the low particle size samples (BLE32-BLE125), the amorphous particles had higher nitrogen (N) and oxygen (O) content compared to the fibrous particles (Table 3). However, in the larger particle size samples (BLE200-BLE500), this difference was not observed (Table 3).

Regarding the chemical composition of the fibrous component, it was observed that the samples exhibiting high particle size and high fibrous content (BLE200-BLE500) demonstrated higher concentrations of N and O groups (Table 3).

#### 3.1.2. Characterisation of the (BLE) Particles by FTIR-ATR Spectroscopy

The analysis of the original (BLE) particles, as well as the investigation of the impact of the particle size distribution on the material’s functional groups, was conducted through the utilisation of FTIR-ATR spectroscopy. In Figure 5, the spectra of BLE32, BLE63, BLE125, BLE200, and BLE500 are presented for comparison.

The most abundant compounds in both materials were cellulose, hemicelluloses and lignin, as shown by the intensity of the vibration bands in the (BLE) spectra. However, other low-molecular-weight sugars and polyphenolic compounds were also detected. The FTIR bands of these materials were assigned and summarised in Table 4 [24,25,26,27,28,29,30,31]. The FTIR spectra were normalised based on the most intense C-O, C-C and C-C-O stretching vibration bands of cellulose, hemicelluloses, and lignin, which appear at 1025–1035 cm^−1^ [32].

The cellulose content is characterised by the presence of a specific pattern of bands in the FTIR spectra. The most relevant of these are the CH_2_ deformation at 1444 and 1315 cm^−1^, the CH deformation at 1367 cm^−1^, the C-O stretch of C-OH/C-O-C at 1153 cm^−1^, and the CO stretch at 1027 cm^−1^ [31,32]. The area of these characteristic bands were measured, and the potential cellulose content was evaluated (Table 5).

The samples of high particle size distribution (BLE200 and BLE500) exhibited high area percentage bands at 1444 and 1367 cm^−1^ of the CH_2_ and CH cellulosic groups, respectively, while the BLE125 was the sample with the lowest percentage area band at 1027 cm^−1^ related to the C-O, C-C, and C-C-O stretching vibrations of cellulose. Considering all the characteristics of the cellulose band areas, BLE63 and BLE125 were the samples with low cellulose content.

The FTIR band around 1727 cm^−1^ is due to the C=O stretching vibration of carboxyl and acetyl groups, characteristic of the presence of hemicellulose [31,32]. Regarding the intensity of this band on the spectra, the hemicellulose content is generally higher for the high particle size samples.

Lignin is the second most abundant component in BLE by-products. Table 6 shows the important functional groups found in lignin, such as hydroxyl, carbonyl, methoxyl, carboxyl, and aromatic and aliphatic C-H groups. These were detected in the BLE32-500 FTIR spectra based on previous studies [29].

The results showed significant differences between the amounts of lignin in the BLE125 and BLE500 samples.

To complete the evaluation of the (BLE) samples, the presence of oils and waxes was analysed (Table 7). The presence of aliphatic components is indicated by the two vibration bands around 2920 and 2850 cm^−1^, as well as the bands at 1468, 1313, and 725 cm^−1^. These bands are due to vibrations of CH_2_ groups.

The aliphatic character of the (BLE) particles is related to their hydrophobicity and its impact on the polymerisation reaction and rheology of the WPU formulation.

Regarding the results obtained, it was observed that the sample with the smallest particle size (BLE32) had a significantly lower wax content.

### 3.2. Production and Characterisation of Coated Fabrics with (BLE) Particles

#### 3.2.1. (BLE) Particles Impact on WPU Coating Formulation

The first step in the production of the coated textile samples was the preparation of the polymeric solution with the (BLE) particles and to evaluation of the impact of the different (BLE) particle size distributions in the rheological behaviour of the polymeric solutions.

The coating solutions were prepared by mixing the water-soluble polyurethane formulation with 10.0 ± 0.5% (dry basis) of the different (BLE) particles. The coating formulations were characterised in terms of viscosity, foam density, and solid content (Table 8).

In all cases, (BLE) particles increased the viscosity of the polymer formulation. However, no trend was observed with the particle size distribution of the samples.

Due to operational conditions, the coating formulations were foamed and used as stable foam, which made it necessary to evaluate the impact of (BLE) particles on foam density. In general, no foam density increase due to the (BLE) particles was observed, except when BLE32 and BL63 particles were used. This increase could be attributed to the lower proportion of aliphatic compounds present in the particles.

#### 3.2.2. Analysis of Fabrics Coated with (BLE) Particles


Analysis of the Effect of (BLE) Particles on the Colour of Coated Fabrics


The first evaluation was focused on the impact of the (BLE) particles on the esthetical properties of the coated textile products. Figure 6 shows the textile samples coated with WPU formulation and WPU + BLE particles.

The evaluation of colour coordinates was conducted using the CIELab colour measurement system, with the results presented in Table 9.

The (BLE) particles changed the colour of the coated fabrics produced [33]. The effect of the size distribution of the (BLE) particle samples used in the coating formulation on the colour properties of the final product was also confirmed. It was possible to obtain coated fabrics with different colour properties depending on the (BLE) sample.

The coating textile samples with a higher luminescence value L*, indicating a lighter final colour, were those in which large-size particles were used (BLE500, BLE200, and BLE125), which is related to the lower water-extractable components present in the (BLE) samples.

However, textiles coated with the larger particle size samples (BLE200 and BLE500) showed the highest overall colour variation (ΔE*), which is related to a higher proportion of dark spots due to the larger particles present in the (BLE) samples.

Particle size had a greater effect on the colour parameter a* than on b*, with fabrics coated with the smallest (BLE) particle sizes (BLE32-BLE125) having higher a* values (redder).
Colour Lightfastness Evaluation

The lightfastness of the coated cotton samples was then evaluated, which is a relevant parameter that could affect their industrial viability, especially if these products are to be manufactured for demanding applications, such as automotive developments [34,35].

The samples were exposed to artificial white light for one week, after which the colour changes were measured (Figure 7).

The most relevant difference in the colour of the coated fabrics due to the light treatment was observed in the parameters L* (lightness) and b* (yellow blue). The light treatment increased the L* value of all coated fabrics (brighter colour), with this increase higher in the fabrics coated with (BLE) particles. However, due to the size distribution of the (BLE) particles used in the coating formulation, no significant trend in the increase of the L* value was observed. As for the b* parameter, exposure to light in all samples changed the colours of the coated fabrics to lower b* values, resulting in bluer colours. The fabrics coated with the largest (BLE) particle size (BLE125-BLE500) had significantly higher Δb* values due to light treatment.

Colour lightfastness can be measured by ΔE* (total colour difference) values, which are obtained by comparing the parameters of the colour values of the coated fabric before and after exposure to light [6]. The (BLE) particles influenced the colour lightfastness of the coated fabrics, and the influence of the size distribution of the (BLE) particles used was also significant. The coated fabrics produced with the smallest (BLE) particle size had the lowest ΔE* values and the highest colour lightfastness. This is related to the higher antioxidant capacity of the water extracts obtained from the samples with the smaller particle size distribution (BLE32, BLE63), values analysed in a previous work of this research group [6].
Characterisation of the Samples by SEM Microscopy

Scanning electron microscopy (SEM) was used to evaluate the morphology of coated fabrics produced without (PB) and with (BLE) particles (Figure 8).

The most significant differences detected due to the presence of (BLE) particles are related to the percentage of foam voids present in the polymer matrix. The water-based WPU formulation was applied as a foam and then hot pressed in a calender press to achieve the required end thickness.

The (BLE) particles altered the coating layer, producing a textile product with a higher proportion of retained gas and surface voids.

The voids in the surface and in the polymer, matrix will affect several parameters that will influence the final applicability of the products, namely wettability, breathability and thermal transfer properties. The coatings prepared with BLE63 were the ones that achieved a higher proportion of voids on the surface and bubbles retained in the polymer matrix.
FTIR-ATR Coating Samples Characterisation

The fabrics coated with the WPU formulation alone (PB) and with the different particle sizes (BLE) (BLE32-500) were analysed by FTIR-ATR spectroscopy, and the absorbance spectra are shown in Figure 9.

The FTIR bands of the coated fabrics were assigned and summarised in Table 10. The FTIR spectra were normalised on the basis of the characteristic -C-O-C- stretching peak band of the polyol component, which appears at 1157 ± 5 cm^−1^, refs. [36,37] and the area of each characteristic peak was calculated as a percentage (relative to the total area of the spectra). The experiments were performed in triplicate at different points and the results were averaged.

No isocyanate absorption band at 2266 cm^−1^ was observed in any of the samples analysed, indicating that the polyurethane curing reaction was complete in all cases and that no free isocyanate groups were present in the coated fabrics [41].

Concerning to the characteristic polyurethane bands, the presence of an N-H vibration band at 3383 cm^−1^ [41], the carbonyl stretching non-hydrogen bonded urethane band at 1724 cm^−1^ and the associated urethane (hydrogen bonded) band at 1687 cm^−1^ were observed, the amide II band at 1518 cm^−1^ due to the stretching vibration of the C-N bonds and the deformation vibration of the C-N-H bonds of the urethane linkages, and the amide III band at 1245 cm^−1^ due to the deformation vibration of the N-H bond and the deformation vibration of the O-C-N bonds.

The fabrics coated with the formulation containing the (BLE) particles showed a significant reduction in the area of the bands at 1518 cm^−1^ (of amide II), regardless of the particle size used, with the BLE125 being the sample with the lowest value.

Regarding the band at 1245 cm^−1^ (of amide III), only the samples produced with BLE63 and BLE125 particles showed a significant reduction in the area of the band compared to the reference coated fabric (PB).

The bands of hydrogen-bonded and non-hydrogen-bonded carbonyl groups appear at 1685 and 1724 cm^−1^, respectively. The fabrics coated with BLE63, BLE125 and BLE200 particles showed a smaller band area due to the non-hydrogen bonded urethane carbonyl groups (PB), but only BLE 125 showed a significant reduction in the band due to hydrogen-bonded urethane bonds.

In general, the area of the CH_2_ and CH_3_ bands was smaller in fabrics coated with (BLE) particles than in fabrics coated only with WPU (PB).

Finally, changes were observed in the OH band that appeared at 3520 cm^−1^, with the area increasing in the fabrics coated with the (BLE) particles, the highest values being obtained when BLE32 and BLE500 particles were used. The hydroxyl groups present on the coating surface will affect the reactivity and hydrophobicity of the material.
Hydrophobicity and Oleophobicity of Coating Textile with (BLE)

The hydrophobicity and oleophobicity of the coated fabrics produced with the different (BLE) particle samples (BLE32-500) and solely with the WPU formulation (PB) were characterised by measuring the contact angle of a drop of water and hexadecane on the surface of the coating (Figure 10). These properties will influence the applicability of the products in product manufacturing, their cleanability, disinfectability, breathability, and impermeability [42,43].

In all textile products coated with (BLE) particles, the hydrophobicity of the coated surface was lower than the reference (PB), which correlates with the higher hydroxyl surface groups observed by FTIR-ATR (Table 10).

In terms of oleophobicity, all the samples coated with the (BLE) particles showed a significant reduction in oleophobicity, with the fabrics coated with the BLE63 and BLE125 particles exhibiting low contact angle values. Both coatings were the ones that showed low FTIR area percentage values of the bands at 1724 and 1245 cm^−1^ due to the non-hydrogen bonded urethane and amide III vibration of the urethane bonds, respectively (Table 10).
Coating Textile with (BLE) Particles Performance

The effect of the (BLE) particles on the physico-mechanical behaviour of the coated fabric was evaluated and the results are shown in Figure 11 and Figure 12. The original cotton fabric used as a textile substrate was also tested as a reference (MAM).

In all cases, the coating increased the elasticity and tensile strength of the textile product. To analyse the effect of the particles on the coating, the behaviour of the material when subjected to low forces (<50 N) was analysed measuring the elongation value, which is a parameter of great interest for the applicability of the coating on textile products, as it is related to the reversible deformation that the material can withstand. In addition, maximum force and elongation were also tested.

Fabrics coated with BLE32 and BLE125 particles showed the lowest elongation at 50 N, while the most elastic textile was coated with the largest particle size (BLE500) and without the addition of (BLE) particles (PB). This higher elasticity of the product coated with the largest (BLE) particles is related to the higher percentage of fibrous particles and the lower content of extractable phenolic compounds.

Regarding the tensile strength and maximum elongation of the coated textile products (Figure 12), it was observed that regardless of the (BLE) particle size used, the (BLE) particles have a reinforcing effect on the material, increasing the tensile strength value without significantly reducing its elongation capacity. The material coated with the largest (BLE) particles had the highest tensile strength and elongation.
Coating Textile Products Thermal Behaviour Characterisation

DSC Essay

To evaluate the effect of (BLE) particles on the thermal behaviour of textile products coated with them, a DSC test was carried out in an N_2_ atmosphere. The thermogram, along with the characteristic temperatures and enthalpies of the thermal process, is shown in Figure 13 and Table 11.

Two endothermic peaks associated with thermal events in the cotton knit were observed: the first at around 70 °C, associated with the moisture, and the second at around 370 °C, related to the formation of volatile compounds during the decomposition of cellulose [44].

Concerning to the effect of the coating on the thermal behaviour of the fabric, it was observed that the first endothermic peak shifted to higher temperature values (79–94 °C), because at this temperature the melting of the hard segments of the polyurethane matrix also occurs [45], and also a new peak appears around 220–227 °C, related to the melting of the soft segments of the polyurethanes [46].

As for the degradation peaks, a peak at 355–362 °C was detected, which overlaps with the degradation peak of the textile substrate, and a second endothermic degradation peak at 451–457 °C, which is not present in the textile substrate.

The (BLE) particles had shifted the first peak (79–94 °C) to lower temperature values, indicating an effect on the melting of the hard segments of the polyurethane matrix.

Regarding the second peak (220–227 °C) related to the melting of the soft segments, the peak temperature was generally similar for the fabrics coated with (BLE) particles and no differences in peak enthalpy were detected.

The temperature of the peak observed at 356–362 °C, associated with polyurethane degradation, was higher for all fabrics coated with the (BLE) particles compared to the reference (PB). In addition, higher degradation enthalpy values were obtained for the fabrics coated with (BLE) particles, indicating that more energy was required to break the urethane bonds. Finally, a peak was observed at 451–457 °C, which is associated with the onset of degradation of the phenolic structures present in the polyurethane polymer. It was observed that the peak enthalpy was higher for the samples coated with the low (BLE) particle size (BLE32-125), which is related to their increased phenol content [6].
Simultaneous Thermal Analysis (STA)

A STA analysis in O_2_ atmosphere involving TGA and DSC analysis was performed to assess the (BLE) particles’ impact on the thermal stability of the coated textile products. The TGA, DTG and DSC curves are shown in Figure 14. As a reference, the cotton knit used as a textile substrate was along evaluated, along with the textile sample coated only with the WPU formulation (PB).

Thus, it was possible to identify one degradation step on the textile substrate and three degradation steps in the coated textile products. The maximum decomposition temperatures of the different degradation steps and the weight residue at 500 °C are shown in Table 12.

For the coated samples, the first weight loss between (330.5–340.5) was attributed to the degradation of urethane bonds to form primary amine and olefin or secondary amine and carbon dioxide [47]. The second weight loss around 449.4–471.0 corresponds to the structural decomposition of organic chains (mainly urea groups) [18]. In general, the (BLE) particles shift the peak to lower values due to the thermal degradation of the (BLE) particles.

The Td 5%, which is related to the melting of the soft segments and the onset of WPU degradation, was influenced by the type of (BLE) particle sample used in the coating formulation, with this temperature being higher for fabrics coated with BLE63, BLE200, and BLE500 particles.

Td 20% was selected as a reference of the start of the most relevant degradation step, with Td 50% and Td 75% correlated to the middle and the end of the degradation peak. No difference was observed in the Td 20% of the coated fabrics due to the (BLE) particles, being higher the impact of the (BLE) particles on the Td 50% that was low for the fabrics coated with the high particle size distribution (BLE125-500) and mainly in the Td 75%, which was higher for the fabrics coated with the BLE63 and BLE200 particles. In general, the fabric samples coated with the (BLE) particles showed a residual mass at 600 °C higher than the reference coated only with the WPU formulation (PB), with the fabrics coated with the low (BLE) particle size samples BLE32, and BLE63 exhibiting high residual mass values (7.4 and 10.9). This is related to the high phenolic content of the low particle size samples demonstrated in previous works [6].

The results obtained by simultaneous DSC analysis under O_2_ atmosphere were summarised and shown in Table 13.

In the thermograms obtained by DSC, three peaks were observed for the fabric coated only with the WPU formulation, while only two peaks for the coated fabrics in which (BLE) particles were added to the coating formulation, confirming the influence of the (BLE) particles on the properties of the coated fabrics. All the peaks observed in the DSC analysis in O_2_ atmosphere were exothermic peaks, in contrast to the previous tests conducted in N_2_ atmosphere.

The first peak related to the urethane bond degradation appears at lower temperature (338.2–355.6 °C) than that obtained in the previous essays in N_2_ atmosphere (355.9–361.6 °C), demonstrating the O_2_ impact on the urethane degradation; however, the second degradation peak shifted to higher temperature values compared to the tests conducted in O_2_ atmosphere.
Antioxidant Activity of Coating Textiles with (BLE)

The ABTS and FRAP methods were used to evaluate the antioxidant activity of the textile samples. The FRAP method measures the reduction of ferric ion (Fe^3+^) to ferrous ion (Fe^2+^) in the presence of antioxidants [48], while the 2,2-diphenylpicrylhydrazyl (DPPH) radical scavenging capacity method measures the ability of antioxidant compounds to scavenge the DPPH radical [49].

Recent work by the research group has demonstrated the feasibility of producing textile products dyed with (BLE) extracts with antioxidant properties. Following the same strategy, the antioxidant properties of the cotton knitted fabric used as substrate and the textile products coated with the WPU base formulation and with the WPU and BLE32-500 particles were evaluated by DPPH and FRAP methodology, and the results obtained are shown in Figure 15 and Figure 16.

It has been shown that it is possible to obtain functionalised coated textile products with antioxidant properties solely by incorporating (BLE) particles in the textile coating formulation. Regarding the influence of the particle size distribution, it was observed that after 6 h, the coated textile products with the smaller particle sizes (BLE32 and BLE63) were the ones that presented a higher antioxidant activity, which correlated with the higher antioxidant capacity of the extractable compounds present in the samples [6].

As expected, the incubation time increased the antioxidant efficacy for all coated fabrics tested [50].

Finally, the antioxidant capacity of the textile samples was evaluated by FRAP methodology. The coated fabrics were subjected to an ultrasound-assisted alkaline extraction process and the extracts obtained were evaluated by FRAP methodology.

The results obtained are shown in Figure 16.

The results obtained with the FRAP method confirm the DPPH analyses, which show that textiles coated with (BLE) particles possesses antioxidant properties. It has also been shown that the textiles coated with the smallest particle sizes (BLE32-BLE63) have the highest antioxidant activity, which is related to the characterisation of extractable compounds present in the samples carried out in a previous work by the research group [6].

## 4. Conclusions

The study demonstrates the potential of using unseparated small branches and leaves as a component of textile coatings. Replacing 10% of the polyurethane with forest by-products resulted in a new product with differential colour, feel, and mechanical performance. The particles improved the tensile strength of the coated fabrics without reducing their elongation, making them suitable for high-performance footwear. The thermal properties of the coated fabrics were also evaluated, showing that the addition of BLE particles increased the residual mass at 600 °C by more than four times. The transfer of the antioxidant properties of the eucalyptus particles to the textile product was confirmed, providing a functional textile product with potential applications in the medical, pharmaceutical, and sports sectors. Further research could explore the economic and environmental benefits of using forest by-products in textile manufacturing. Overall, the study highlights the potential of using forest by-products in developing new functional textile products.

## Figures and Tables

**Figure 1 polymers-17-01619-f001:**
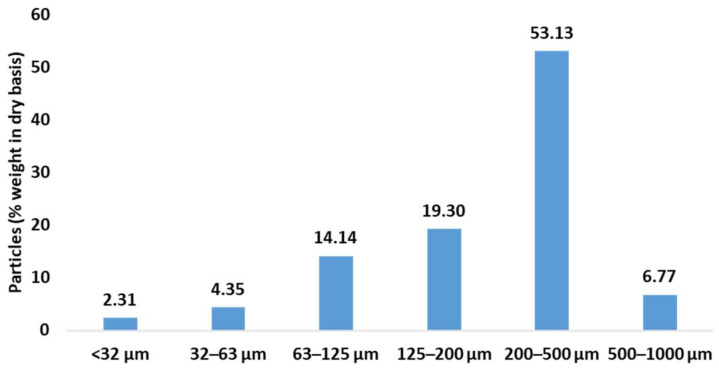
Particle size distribution in weight of the original milled (BLE) particles.

**Figure 2 polymers-17-01619-f002:**
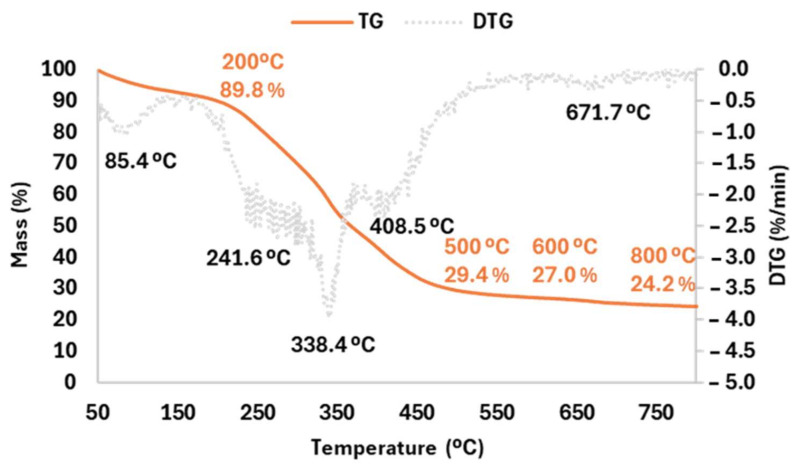
Thermogravimetric analysis curve (TGA) and first derivative (DTG) obtained for Eucalyptus (BLE) particles.

**Figure 3 polymers-17-01619-f003:**
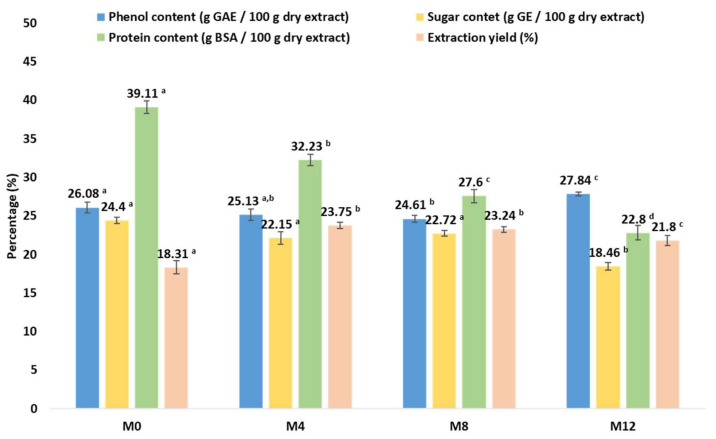
Evaluation of the storage time impact on the (BLE) particles chemical composition (extraction yield, protein, sugar, and phenol content). Samples marked with different letters exhibit significant differences in storage time impact *p* < 0.05.

**Figure 4 polymers-17-01619-f004:**
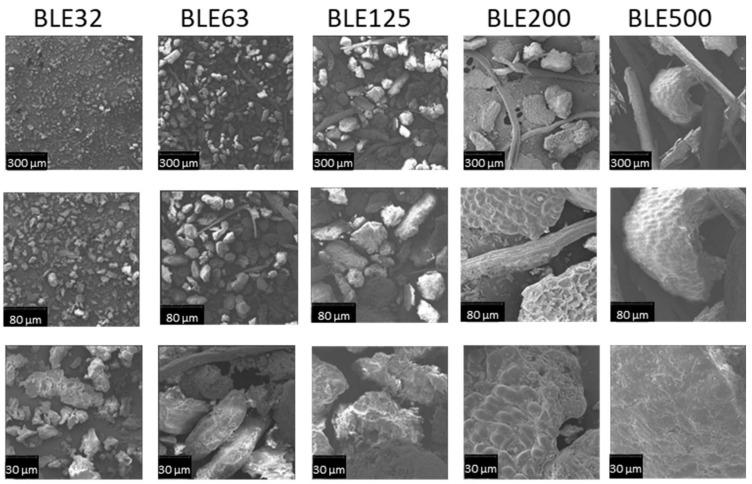
SEM micrographs of the (BLE) samples (BLE32, BLE125 BLE200, and BLE500) at 250, 1000, and 2000 magnifications.

**Figure 5 polymers-17-01619-f005:**
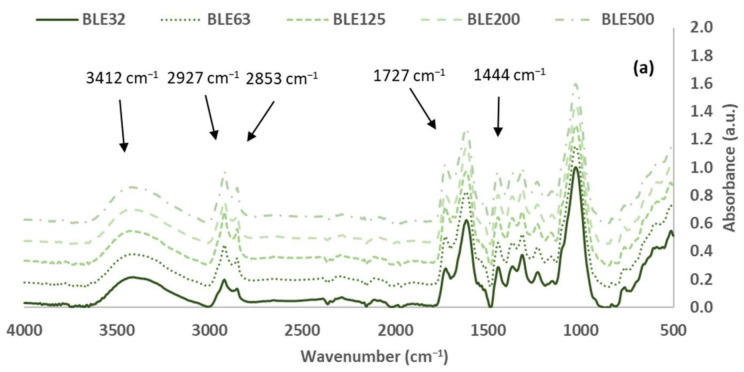
FTIR spectra: *Eucalyptus Globulus* branches and leaf particles (BLE) with different particle size distributions (**a**) (500–4000 cm^−1^) and (**b**) (600–1800 cm^−1^).

**Figure 6 polymers-17-01619-f006:**
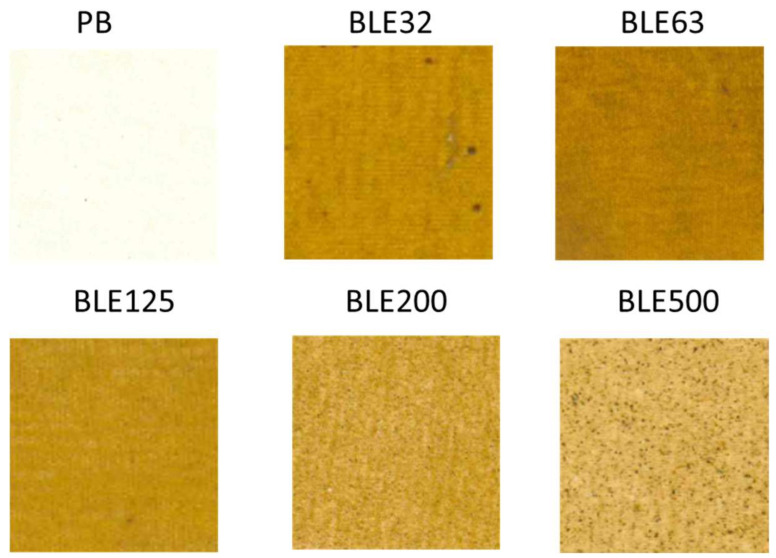
Coated fabrics with aqueous WPU formulation (PB) and with WPU + BLE32-BLE500 particles (BLE32-BLE500).

**Figure 7 polymers-17-01619-f007:**
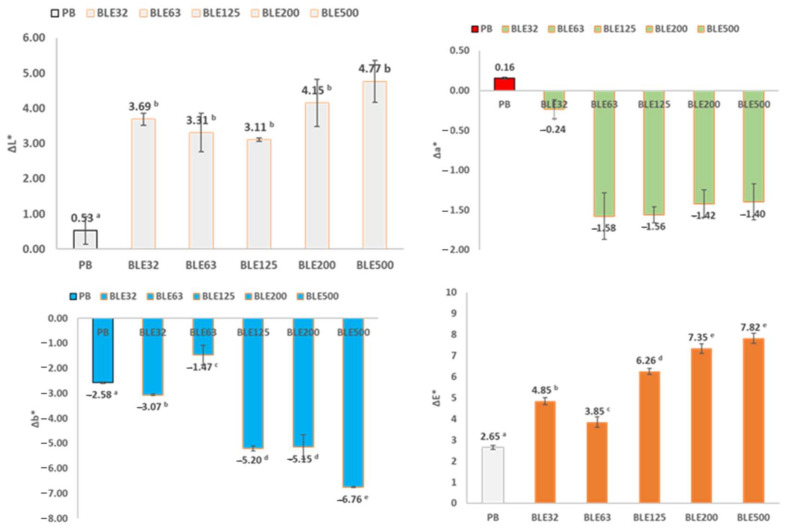
Shows the colour changes of the coated fabrics, in the CIELAB colour space due to the light exposure. (ΔL*, Δa*, Δb*, and ΔE* are the variation of the CIELAB colour space parameters) Samples marked with different letters exhibit significant differences in terms of the impact of raw material particle size *p* < 0.05.

**Figure 8 polymers-17-01619-f008:**
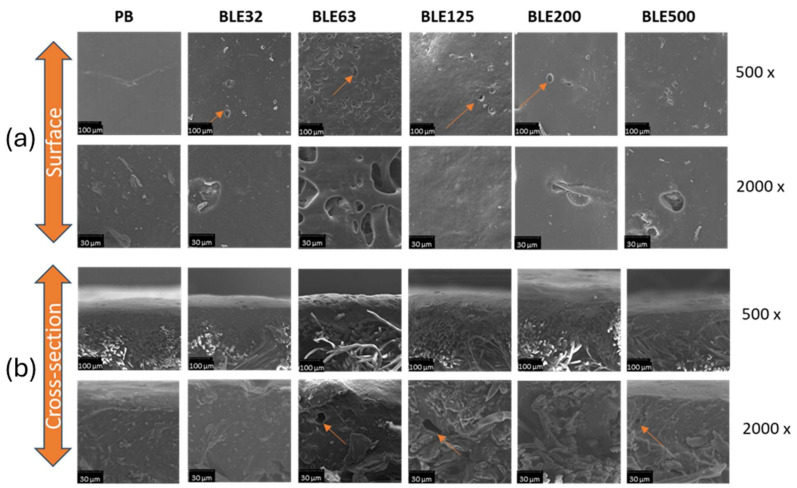
Scanning electron microscopy micrographs of the coated textile samples produced with the WPU solution alone and mixed with the (BLE) particles surface (**a**) and cross section (**b**) at 500 and 2000 magnifications.

**Figure 9 polymers-17-01619-f009:**
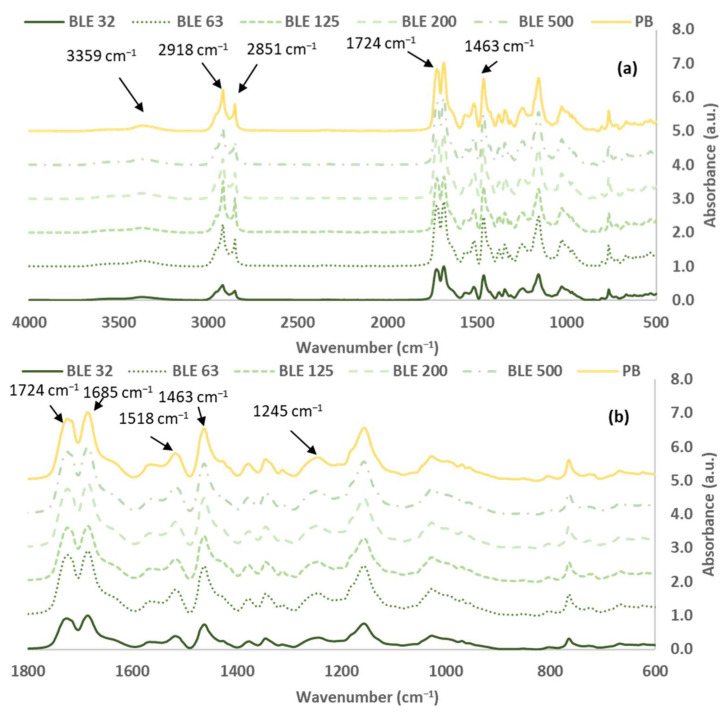
FTIR spectra: Coatings without (PB) and with branches and leaves of *Eucalyptus Globulus* (BLE) featuring different particle size distributions (**a**) (500–4000 cm^−1^) and (**b**) (600–1800 cm^−1^).

**Figure 10 polymers-17-01619-f010:**
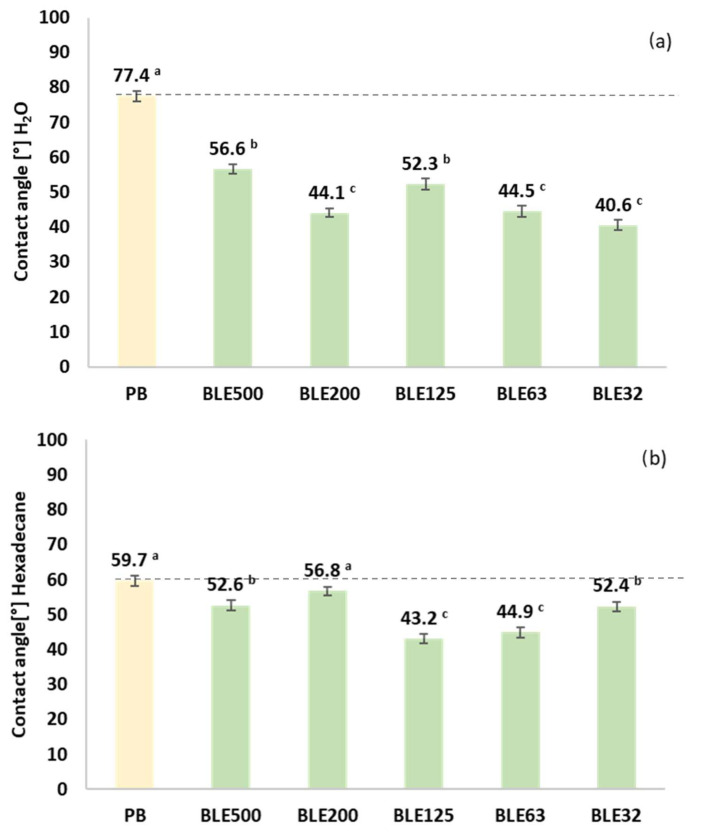
Wettability of coated textile products without (PB) and with branches and leaves of Eucalyptus (BLE) with different particle size distributions (**a**) water contact angle and (**b**) hexadecane contact angle Values are presented as mean (n = 6). Samples marked with different letters exhibit significant differences in terms of the impact of raw material particle size *p* < 0.05.

**Figure 11 polymers-17-01619-f011:**
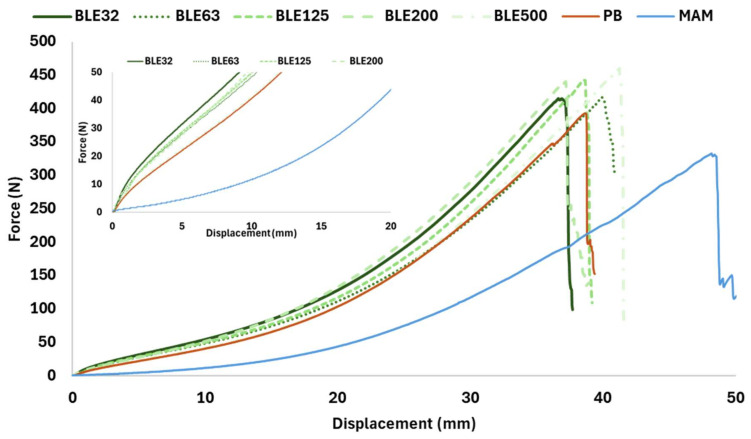
Force versus elongation in the wale direction of the original cotton fabric (MAM) and the coated without (PB) and with (BLE) particles (BLE32-500) (sample dimensions: 4 × 10 cm; speed 20 mm/min).

**Figure 12 polymers-17-01619-f012:**
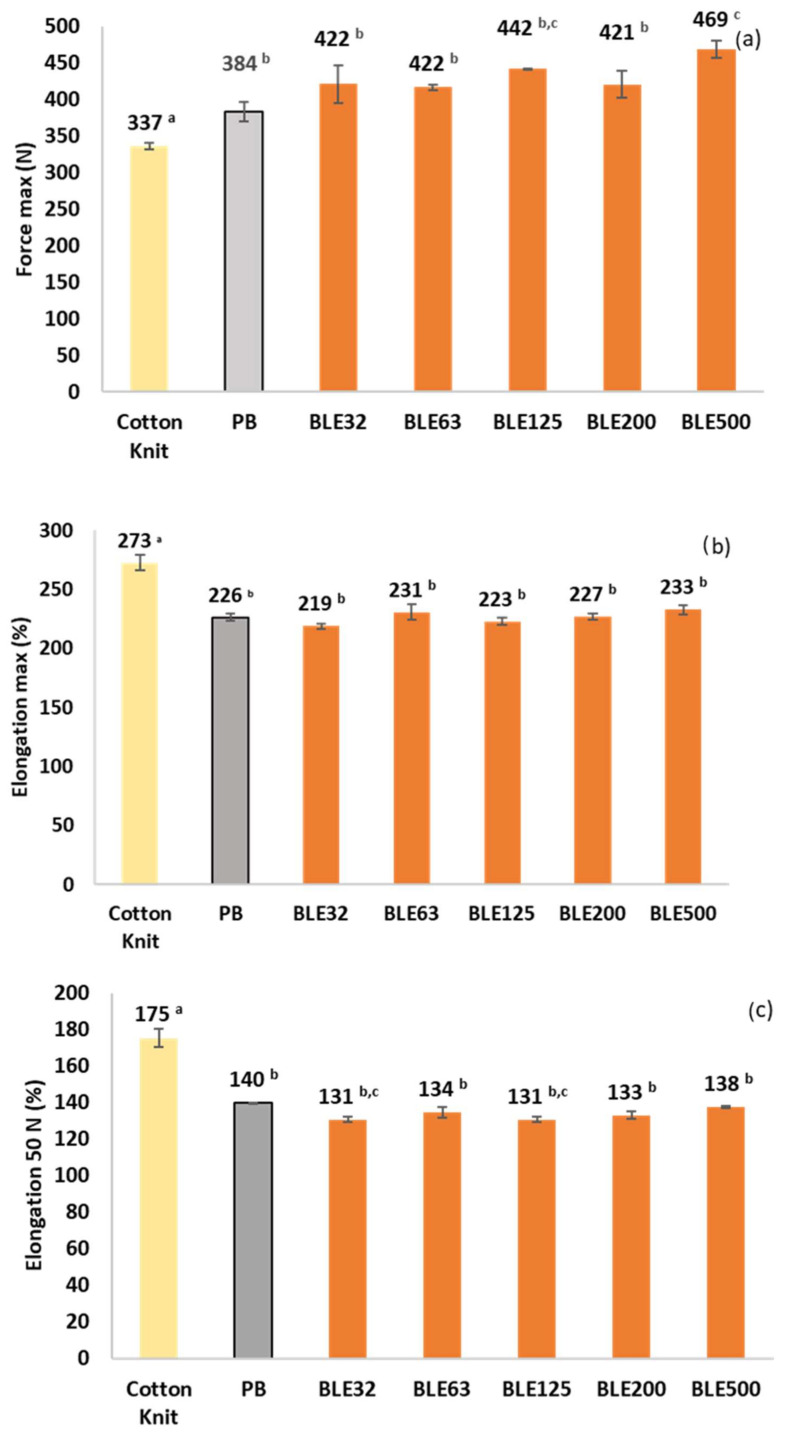
Breaking forces (N) (**a**), maximum elongation (%) (**b**), and elongation at 50 N (**c**) measured in the test samples in the wale direction of the coated cotton knits (samples dimensions: 4 × 10 cm; speed: 20 mm/min). Samples marked with different letters exhibit significant differences in terms of impact of raw material particle size *p* < 0.05.

**Figure 13 polymers-17-01619-f013:**
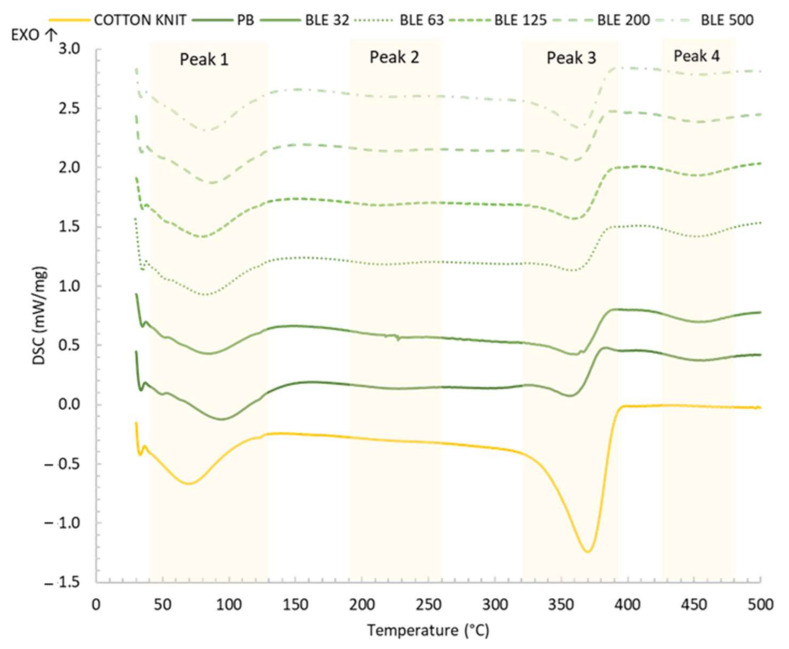
DSC curves of cotton knit and coated with WPU alone (PB) and with the (BLE) particles (BLE32-500) and at a heating rate of 10 C min^−1^ in N_2_ atmosphere.

**Figure 14 polymers-17-01619-f014:**
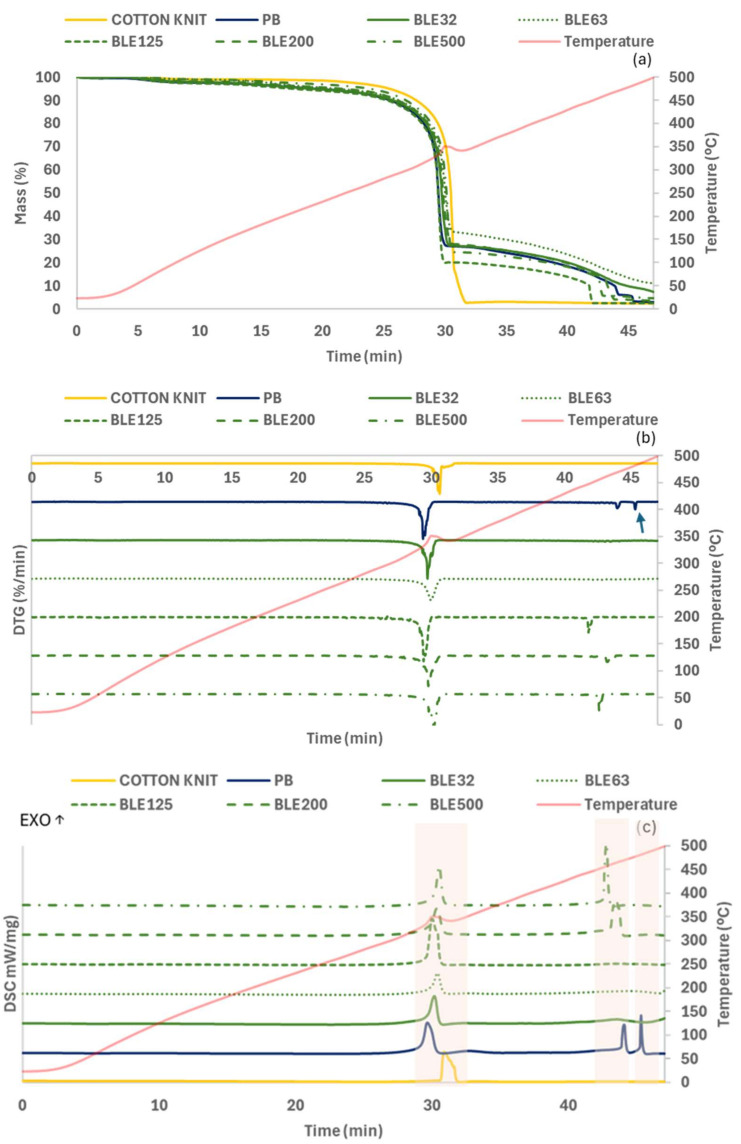
TG (**a**), DTG (**b**) and DSC (**c**) curves of PB and BL E32-500 coated textile products. Note: (**b**) graph shows the first derivatives (DTG) illustrating the decomposition temperatures (Td) of the textile products.

**Figure 15 polymers-17-01619-f015:**
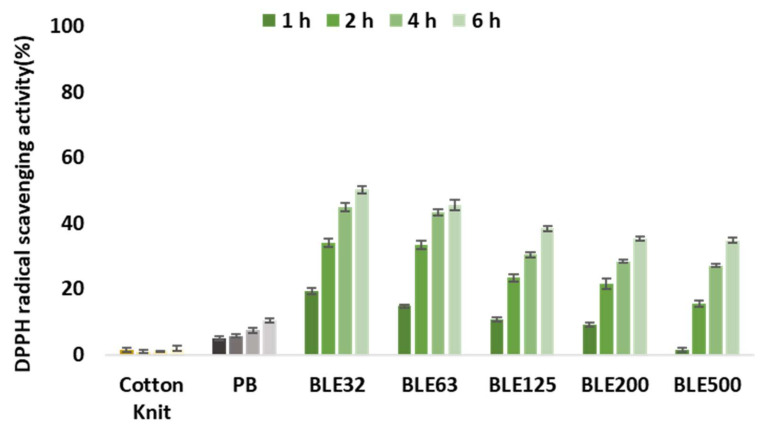
The ‘DPPH radical scavenging activity (%)’ at different times of cotton fabrics uncoated and coated with the WPU formulation alone (PB) and with the (BLE) particles (BLE32-500).

**Figure 16 polymers-17-01619-f016:**
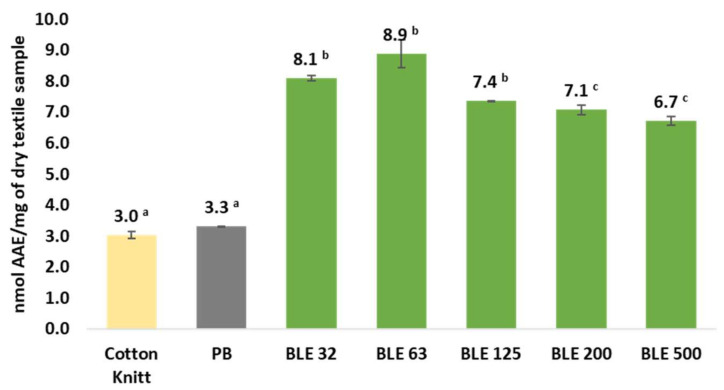
The FRAP antioxidant capacity of cotton fabrics uncoated and coated with the WPU formulation alone (PB) and with the (BLE) particles (BLE32-500). Samples subscripted by different capital letters are significantly different at *p* < 0.05.

**Table 1 polymers-17-01619-t001:** Composition of coating formulations without (PB) and with BSG particles.

Sample	Polyurethane % (wt/wt Dry Basis)	Isocyanate % (wt/wt Dry Basis)	(BLE) % (wt/wt Dry Basis)	Additives % (wt/wt Dry Basis)
PB	62.0	29.5	0.0	8.6
BLE32	55.8	26.5	10.0	7.7
BLE63	55.8	26.5	10.0	7.7
BLE125	55.8	26.5	10.0	7.7
BLE200	55.8	26.5	10.0	7.7
BLE500	55.8	26.5	10.0	7.7

**Table 2 polymers-17-01619-t002:** Dimension analysis of the (BLE) particles.

Samples	Particle Size (µm)	Fiber		Amorphous Particle
		Length (µm)	Width (µm)	Length (µm)
BLE32	>32	247.0 ± 41.0	15.0 ± 2.1	28.6 ± 11.9
BLE63	32–63	225.9 ± 83.2	18.7 ± 5.3	51.2 ± 14.4
BLE125	63–125	535.8 ± 180.2	45.0 ± 4.2	112.5 ± 39.9
BLE200	125–200	176.6 ± 43.2	49.3 ± 7.3	176.6 ± 43.2
BLE500	200–500	912.5 ± 20.1	136.5 ± 7.8	398.0 ± 70.7

**Table 3 polymers-17-01619-t003:** Chemical characterisation of the (BLE) particles surface by EDS.

Samples	Fiber			Amorphous Particle		
	C (%)	O (%)	N (%)	C (%)	O (%)	N (%)
BLE32	74.7 ± 2.9 ^a^	21.8 ± 2.5 ^a^	3.6 ± 1.4 ^a^	66.0 ± 6.9 ^a^	26.5 ± 6.4 ^a^	7.5 ± 2.6 ^a^
BLE63	66.0 ± 5.5 ^b^	27.9 ± 4.8 ^b^	6.1 ± 0.8 ^b^	52.2 ± 5.9 ^a^	32.9 ± 8.5 ^a^	14.9 ± 3.6 ^b^
BLE125	65.7± 6.2 ^b^	28.9 ± 4.9 ^b^	6.2 ± 0.6 ^b^	60.6 ± 7.8 ^a^	31.5 ± 7.6 ^a^	7.8 ± 1.6 ^a^
BLE200	48.3 ± 0.9 ^c^	42.3 ± 2.8 ^c^	10.8 ± 2.0 ^b^	62.1 ± 3.1 ^a^	30.8 ± 3.5 ^a^	7.0 ± 2.2 ^a^
BLE500	50.7 ± 5.6 ^c^	40.5 ± 4.1 ^c^	10.0 ± 3.6 ^b^	47.2 ± 5.8 ^b^	44.5 ± 6.4 ^b^	8.2 ± 4.0 ^a^

Values are presented as mean ± standard deviation (n = 12). (%) were calculated as weight concentration; samples marked with different letters exhibit significant differences in terms of impact of raw material particle size *p* < 0.05.

**Table 4 polymers-17-01619-t004:** FTIR peak assignment of (BLE) samples.

	BLE32	BLE63	BLE125	BLE200	BLE500		
cm^−1^	Intensity	Intensity	Intensity	Intensity	Intensity	Group	Range
3412	21.7 ± 0.1	23.3 ± 0.2	24.6 ± 0.2	25.2 ± 0.7	26.0 ± 0.7	-OH stretch	3300–3400
2920	19.9 ± 0.3	29.9 ± 2.9	39.0 ± 3.3	30.8 ± 3.3	37.7 ± 7.0	-CH_2_- asymmetric stretch	2916–2936
2853	13.5 ± 0.2	20.4 ± 2.0	26.5 ± 2.3	20.7 ± 2.3	25.7 ± 4.9	-CH_2_- symmetric stretch	2843–2863
1727	27.9 ± 0.7	35.8 ± 1.9	42.9 ± 3.4	36.6 ± 2.1	42.5 ± 3.6	C=O stretch in unconjugated ketones, carbonyls and in ester groups (hemicellulose)	1738
1615	62.3 ± 1.6	67.6 ± 0.5	67.2 ± 1.8	69.8 ± 0.6	68.5 ± 4.8	Aromatic skeletal vibration and C=O stretch (lignin)	1595
1551	17.7 ± 0.4	19.8 ± 0.0	17.9 ± 1.0	17.5 ± 0.9	17.6 ± 0.8	C_AR_ = C_AR_ (Pp cd.)	1500–1600
1454	23.7 ± 0.8	27.0 ± 1.2	30.5 ± 1.4	31.1 ± 1.5	33.6 ± 3.2	C=C and C-H bond O-H in plane deformation (lignin and hemicellulose)	1450–1453
1444	28.8 ± 0.2	31.3 ± 0.4	34.7 ± 0.6	35.4 ± 1.0	37.0 ± 3.8	CH- deformation; asymmetric in -CH_3_ and -CH_2_- (cellulose)	1430–1485
1367	29.4 ± 0.2	31.5 ± 0.2	34.1 ± 0.7	34.4 ± 0.7	36.1 ± 2.6	CH deformation (cellulose and hemicellulose)	1372
1315	37.5 ± 0.6	37.4 ± 0.3	38.3 ± 0.4	40.0 ± 0.6	39.7 ± 2.9	Ph-CHR-OH deformation	1260–1350
1232	25.3 ± 0.1	28.2 ± 0.2	29.9 ± 1.4	28.6 ± 1.5	29.9 ± 2.1	Syringyl ring and C=C stretch in lignin and xylan	1235
1153	18.9 ± 0.0	21.5 ± 0.9	24.0 ± 2.2	21.1 ± 0.8	23.2 ± 1.8	Involves C-O stretching of C-OH/C-O-C (cellulose)	1160
1027	100.0 ± 0.0	100.0 ± 0.0	100.0 ± 0.0	100.1 ± 0.1	100.0 ± 0.1	C-O, C-C, and C-C-O stretch (cellulose, hemicellulose, and lignin)	1025–1035
832	2.3 ± 0.1	2.6 ± 0.1	3.0 ± 0.1	3.4 ± 0.3	3.6 ± 0.4	C-O-C aromatic ethers, symmetric stretch	810–850
763	14.4 ± 0.3	13.6 ± 0.0	12.7 ± 0.6	12.4 ± 1.3	12.9 ± 0.7	C-C Alkanes skeletal vibrations	720–750

Ph: phenyl group; BLE_E_: milled and sieved branches and leaves of *E. globulus*.

**Table 5 polymers-17-01619-t005:** Cellulose FTIR bands quantification for (BLE) particles.

	BLE32	BLE63	BLE125	BLE200	BLE500	Group	Integration Range
cm^−1^	% Area	% Area	% Area	% Area	% Area		
1444	3.12 ± 0.05 ^a^	3.09 ± 0.01 ^a^	3.40 ± 0.00 ^a^	3.47 ± 0.00 ^b^	3.53 ± 0.00 ^b^	CH_2_ deformation medium-weak	1400–1485
1367	2.18 ± 0.02 ^a^	2.16 ± 0.02 ^a^	2.24 ± 0.00 ^a^	2.32 ± 0.02 ^b^	2.35 ± 0.05 ^b^	CH deformation	1390–1350
1315	3.62 ± 0.05	3.36 ± 0.01	3.29 ± 0.01	3.54 ± 0.09	3.35 ± 0.12	CH_2_ deformation medium-weak	1340–1280
1153	2.15 ± 0.04	2.20 ± 0.13	2.26 ± 0.23	2.09 ± 0.06	2.20 ± 0.01	C-O stretch of C-OH/C-O-C medium	1195–1130
1100	2.72 ± 0.16	2.51 ± 0.17	2.44 ± 0.17	2.52 ± 0.14	2.49 ± 0.15	C-O-C stretch medium	1130–1090
1027	13.55 ± 0.09 ^a^	12.52 ± 0.03 ^a^	12.05 ± 0.20 ^b^	12.60 ± 0.06 ^a^	12.40 ± 0.62 ^a^	CO stretch medium-strong	1068–990
∑ Area	27.16 ± 0.21 ^a^	25.75 ± 0.31 ^b^	25.59 ± 0.14 ^b^	26.45 ± 0.16 ^a^	26.29 ± 0.61 ^a^		

BLE_32-500_: milled and sieved branches and leaves of *E. globulus*; % area was the peak area/area total; area total was measured between 400 and 4000 cm^−1^. Values are presented as mean ± standard deviation (n = 3); samples marked with different letters exhibit significant differences in terms of the impact of raw material particle size *p* < 0.05.

**Table 6 polymers-17-01619-t006:** Lignin FTIR bands quantification for (BLE) particles.

	BLE32	BLE63	BLE125	BLE200	BLE500	Group	Integration Range
cm^−1^	% Area	% Area	% Area	% Area	% Area		
1688	1.72 ± 0.32	1.74 ± 0.09	1.86 ± 0.23	1.68 ± 0.14	1.60 ± 0.23	C=O stretch	1670–1700
1616	7.61 ± 0.51	7.87 ± 0.30	7.60 ± 0.36	7.70 ± 0.52	7.33 ± 0.71	Aryl ring stretch, asymmetric	1560–1640
1516	0.62 ± 0.03 ^a^	0.69 ± 0.01 ^b^	0.58 ± 0.04 ^a^	0.57 ± 0.01 ^a^	0.57 ± 0.01 ^a^	Aryl ring stretch, asymmetric	1488–1525
1446	3.39 ± 0.13	3.50 ± 0.06	3.60 ± 0.02	3.24 ± 0.14	3.43 ± 0.71	OH deformation, asymmetric, OCH_3_ CH deformation, asymmetric, S-mode	1400–1485
1315	3.21 ± 0.26	3.08 ± 0.10	2.99 ± 0.26	3.08 ± 0.28	2.74 ± 0.58	Aryl ring breathing mode; CO stretch; S-mode.	1290–1340
1234	3.13 ± 0.14	3.09 ± 0.19	3.18 ± 0.31	3.07 ± 0.35	3.20 ± 0.36	Syringyl ring and C=C stretch in lignin and Xylan	1195–1265
1160	2.08 ± 0.11	2.16 ± 0.09	2.23 ± 0.03	2.11 ± 0.07	2.14 ± 0.29	C-H stretch in G-ring	1135–1190
∑ Area	20.89 ± 0.14 ^a^	21.32 ± 0.28 ^a^	22.50 ± 0.47 ^a,b^	20.82 ± 0.88 ^a^	19.86 ± 0.57 ^a,a^		

BLE_32-1000_: milled and sieved branches and leaves of *E. globulus*; % area was the peak area/total area; total area was measured between 400 and 4000 cm^−1^. Values are presented as mean ± standard deviation (n = 3). Samples superscripted by different letters are significantly different (*p* < 0.05).

**Table 7 polymers-17-01619-t007:** Aliphatic (waxes) FTIR bands quantification for (BLE) particles.

	BLE32	BLE63	BLE125	BLE200	BLE500	Group	Integration Range
cm^−1^	% Area	% Area	% Area	% Area	% Area		
2920	4.15 ± 0.16 ^a^	5.98 ± 0.19 ^b^	6.35 ± 0.26 ^b^	6.74 ± 0.05 ^b,b^	5.70 ± 0.36 ^b,c^	-CH_2_- asymmetric stretch	3000–2800
2852						-CH_2_- symmetric stretch	
1467	0.41 ± 0.01 ^a^	0.52 ± 0.02 ^b^	0.53 ± 0.03 ^b^	0.53 ± 0.03 ^b^	0.54 ± 0.03 ^b^	CH- deformation; asymmetric in -CH_2_	1480–1460
1315	3.40 ± 0.04	3.11 ± 0.07	3.10 ± 0.02	3.22 ± 0.22	3.24 ± 0.02	CH_2_ deformatio nmedium-weak	1340–1285
721	0.55 ± 0.01	0.58 ± 0.06	0.52 ± 0.04	0.53 ± 0.01	0.55 ± 0.01	C-C Alkanes skeletal vibrations	730–710
∑ Area	8.50 ± 0.22 ^a^	10.35 ± 0.39 ^b^	11.02 ± 0.66 ^b^	11.22 ± 0.39 ^b^	10.03 ± 0.40 ^b^		

BLE_32-1000_: milled and sieved branches and leaves of *E. globulus*; % area was the peak area/area total; area total was measured between 400 and 4000 cm^−1^. Samples marked with different letters exhibit significant differences in terms of impact of raw material particle size *p* < 0.05.

**Table 8 polymers-17-01619-t008:** Characterisation of WPU solutions with (BLE) particles.

Sample	µ (mPa⋅s)	Foam Density (g/L)	Solid Content (%)
PB	8.6	218.1	39.6
BLE32	54.05	263.6	42.5
BLE63	63.41	275.6	42.8
BLE125	70.41	217.1	42.4
BLE200	65.01	222.6	42.2
BLE500	41.25	200.6	43.0

Milled using a sieve of 1000 µm.

**Table 9 polymers-17-01619-t009:** Shows the colour of the coated textile samples, in the CIELAB colour space.

Samples/Coatings	L *	a *	b *	ΔL *	Δa *	Δb *	ΔE *
PB	87.7 ± 0.2 ^a^	0.12 ± 0.10 ^a^	9.2 ± 0.3 ^a^	0	0	0	0
BLE32	51.3 ± 1.6 ^b^	8.10 ± 0.48 ^b^	33.6 ± 1.0 ^b^	−22.3	5.1	23.1	32.5
BLE63	46.8 ± 0.9 ^c^	10.73 ± 0.18 ^c^	30.5 ± 0.9 ^c^	−26.2	6.4	25.3	37.0
BLE125	57.1 ± 0.4 ^d^	7.65 ± 0.15 ^b^	34.9 ± 0.3 ^b^	−30.6	7.5	25.7	40.7
BLE200	61.5 ± 1.0 ^e^	6.54 ± 0.30 ^d^	34.6 ± 0.6 ^b^	−40.9	10.6	21.3	47.3
BLE500	65.4 ± 1.3 ^f^	5.22 ± 0.12 ^e^	32.3 ± 0.4 ^c^	−36.4	8.0	24.3	44.5

Coating conditions: BLE 8% (dry basis). Values are presented as mean ± standard deviation (n = 6). In the CIELAB colour space, * after L, a, and b are part of the full name to distinguish L*a*b* from Hunter’s Lab. Samples marked with different letters exhibit significant differences in terms of impact of raw material particle size *p* < 0.05.

**Table 10 polymers-17-01619-t010:** FTIR peak assignment of coatings without (PB) and with (BLE) particles (PB-BLE).

	PB-BLE32	PB-BLE63	PB-BLE125	PB-BLE200	PB-BLE500	PB	Group	Integration Range (cm^−1^)
cm^−1^	Area (%)	Area (%)	Area (%)	Area (%)	Area (%)	Area (%)		
3520	2.16 ± 0.06 ^a^	2.05 ± 0.16 ^a^	1.87 ± 0.08 ^b^	2.02 ± 0.08 ^a^	2.13 ± 0.04 ^a^	1.22 ± 0.01 ^c^	-OH stretch	3450–3700
3383	4.13 ± 0.09 ^a^	3.66 ± 0.18 ^b^	3.02 ± 0.36 ^c^	3.63 ± 0.09 ^b^	3.75 ± 0.07 ^b^	3.42 ± 0.06 ^b^	-N-H stretch	3200–3450
2917	7.96 ± 0.26 ^a^	8.84 ± 0.17 ^a^	10.1 ± 0.9 ^b^	8.6 ± 0.1 ^a^	8.7 ± 0.5 ^a^	9.1 ± 0.1 ^b^	-CH_2_- asymmetric stretch	2875–3020
2851	2.50 ± 0.10 ^a^	2.96 ± 0.06 ^a^	3.66 ± 0.47 ^b^	2.91 ± 0.04 ^a^	2.88 ± 0.23 ^a^	3.05 ± 0.03 ^a^	-CH_2_- symmetric stretch	2800–2875
--	--	--	--	--	--	--	NCO isocyanate groups stretch	2260–2270
1724	8.37 ± 0.01 ^a^	8.11 ± 0.00 ^b^	7.55 ± 0.05 ^c^	8.03 ± 0.00 ^b^	8.30 ± 0.00 ^a^	8.47 ± 0.00 ^a^	Urethane carbonyl groups non-hydrogen bonded [38]	1745–1705
1685	9.07 ± 0.10 ^a^	8.74 ± 0.02 ^a^	8.08 ± 0.35 ^b^	8.90 ± 0.04 ^a^	9.01 ± 0.14 ^a^	9.19 ± 0.02 ^a^	Urethane carbonyl groups hydrogen bonded [38]	1660–1705
1518	3.55 ± 0.02 ^a^	3.48 ± 0.01 ^a^	3.27 ± 0.04 ^b^	3.47 ± 0.01 ^a^	3.58 ± 0.01 ^a^	3.82 ± 0.00 ^c^	–NH and –C–N vibrations of the urethane linkages [37]	1490–1540
1463	5.75 ± 0.00 ^a^	5.80 ± 0.00 ^a^	5.81 ± 0.01 ^a^	5.98 ± 0.00 ^b^	5.87 ± 0.01 ^a^	6.28 ± 0.00 ^c^	-CH_2,_ -CH_3_ bending vibrations [37,39]	1440–1490
1245	4.84 ± 0.00 ^a^	4.62 ± 0.00 ^b^	4.41 ± 0.01 ^b^	4.70 ± 0.00 ^a^	4.81 ± 0.02 ^a^	4.82 ± 0.00 ^a^	Deformation vibrations of the N-H bond and of the O-C-N bonds	1220–1285
1180	3.67 ± 0.00 ^a^	3.61 ± 0.00 ^a^	3.37 ± 0.03 ^b^	3.64 ± 0.00 ^a^	3.74 ± 0.00 ^a^	3.80 ± 0.00 ^a^	Coupled C-N and C-O stretching vibrations	1165–1195
1065	0.97 ± 0.00 ^a^	0.91 ± 0.00 ^b^	0.91 ± 0.00 ^b^	0.85 ± 0.00 ^c^	0.83 ± 0.00 ^c^	0.76 ± 0.00 ^d^	C–H stretching vibration [40]	1080–1055
1027	3.42 ± 0.01 ^a^	3.34 ± 0.01 ^a^	3.26 ± 0.00 ^a^	3.21 ± 0.00 ^b^	3.14 ± 0.00 ^b^	3.11 ± 0.00 ^b^	C-O, C-C, and C-C-O stretch	1005–1040
764	1.85 ± 0.00 ^a^	1.82 ± 0.00 ^a^	1.71 ± 0.01 ^b^	1.85 ± 0.00 ^a^	1.84 ± 0.01 ^a^	1.67 ± 0.00 ^b^	N-H out of plane bending	740–785
717	0.69 ± 0.00 ^a^	0.74 ± 0.00 ^b^	0.80 ± 0.00 ^c^	0.75 ± 0.00 ^b^	0.74 ± 0.0 ^b^	0.62 ± 0.00 ^d^	C-C Alkanes skeletal vibrations	705–735

PB: textile samples coated with the standard WPU formulation; PB-BLE: textile samples coated with the WPU formulation and the branches and leaves of *E. globulus* particles of different particle sizes. Values are presented as mean ± standard deviation (n = 3). Samples marked with different letters exhibit significant differences in terms of impact of raw material particle size *p* < 0.05.

**Table 11 polymers-17-01619-t011:** DSC characteristic temperatures and enthalpy of the textile products uncoated and coated without (PB) and with the (BLE) particles (BLE32-500).

Samples/Coatings	Peak 1	Peak 2			Peak 3		Peak 4	
	Temperature (°C)	J/g	Temperature (°C)	J/g	Temperature (°C)	J/g	Temperature (°C)	J/g
Cotton Knit	36.9–125.6 (70.2)	−87.8	--	--	326.7–398.0 (369.7)	−220.3	--	--
PB	55.0–125.1 (93.9)	−46.8	175.1–257.0 (227.3)	−7.8	330.7–376.2 (355.9)	−42.2	142.9–494.4 (457.1)	−16.8
BLE32	54.3–138.8 (85.2)	−45.9	180.7–274.6 (227.1)	−7.3	321.5–387.3 (361.6)	−53.0	406.8–499.5 (453.3)	−24.2
BLE63	57.4–124.6 (82.5)	−39.0	170.8–251.0 (218.4)	−8.2	325.1–387.8 (358.8)	−44.7	417.3–499.3 (451.4)	−26.4
BLE125	57.0–120.9 (78.8)	−35.2	162.1–246.9 (211.8)	−8.5	327.7–389.2 (359.2)	−56.1	415.7–497.9 (451.2)	−21.3
BLE200	52.2–125.6 (87.0)	−53.4	162.1–249.7 (221.0)	−7.7	325.6–383.5 (359.7)	−52.5	413.9–498.8 (453.1)	−17.7
BLE500	53.9–122.4 (82.6)	−53.4	162.6–254.8 (220.0)	−8.2	320.3–389.0 (362.5)	−80.8	419.5–494.7 (452.6)	−9.7

Coating conditions: (BLE) 10% (dry basis). Values are presented as mean ± standard deviation (n = 6). Peak temperatures are shown as onset—end of the peak with the value of the peak temperature in brackets.

**Table 12 polymers-17-01619-t012:** DTG (O_2_ atmosphere) characteristic temperatures and TG mass loss values of the textile uncoated and coated products without (PB) and with the (BLE) particles (BLE32-500).

	Cotton Knit	PB	BLE32	BLE63	BLE125	BLE200	BLE500
Peak 1 (°C)	355.7	335.9	340.5	336.1	330.5	330.5	335.0
Peak 2 (°C)	--	471.0	464.3	--	468.2	464.3	456.6
Peak 3 (°C)	--	484.1	--	--	--	--	--
Temperature (Td 5%)	285.7	225.9	221.5	243.3	230.1	246.6	269.0
Temperature (Td 20%)	325.3	315.5	317.8	318.3	313.2	316.8	318.8
Temperature (Td 50%)	345.0	340.6	340.9	337.2	327.8	331.5	333.9
Temperature (Td 75%)	357.5	370.5	379.9	420.6	330.8	381.3	342.4
Residual mass 500 °C (%)	2.4	2.9	7.4	10.9	2.5	3.5	4.7

Td 5%: temperature of 5% weight loss; Td 20%: temperature of 20% weight loss; Td 50%: temperature of 50% weight loss; Td 75%: temperature of 75% weight loss.

**Table 13 polymers-17-01619-t013:** DSC (O_2_ atmosphere) characteristic temperatures and enthalpy of the textile products uncoated and coated without (PB) and with the (BLE) particles (BLE32-500).

Samples/Coatings	Peak 1		Peak 2		Peak 3	
	Temperature (°C)	J/g	Temperature (°C)	J/g	Temperature (°C)	J/g
PB	283.2–357.6 (355.6)	1157	429.2–474.6 (472.9)	597.5	474.6–487.0 (483.4)	267.3
BLE32	279.9–352.4 (350.2)	1028	403.7–483.5 (465.2)	380.8	--	--
BLE63	279.6–347.8 (346.0)	687.9	404.2–495.2 (473.9)	395.9	--	--
BLE125	298.6–342.0 (338.2)	2145	427.0–498.9 (465.0)	288.7	--	--
BLE200	281.3–349.6 (339.6)	1004	427.6–470.5 (469.5)	1147	--	--
BLE500	305.1–345.9 (341.0)	935.4	439.2–463.8 (461.6)	821.2	--	--

Coating conditions: BLE 10% (dry basis). Values are presented as mean ± standard deviation (n = 6). Peak temperatures are shown as onset—end of the peak with the value of the peak temperature in brackets.

## Data Availability

Data is contained within the article.

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
