# Peer review of "New Alternatives in the Valorisation of Eucalyptus globulus By-Products for the Textile Industry"

_polymers, 2025, doi:10.3390/polym17121619_

Round 1
Reviewer 1 Report
Comments and Suggestions for Authors
The authors reported new alternatives in the valorisation of Eucalyptus globulus by-products for the textile industry. This study focuses how BLE particles can be incorporating in textile coating treatment which is of interest for the sustainable textile production. I recommend the publication of this work in Polymers after very minor revisions.
Some questions/comments are listed below:
- The fabric composition is missing in the Raw Material section (Page 3 and Line 133). The authors should also add the fabric's weight (gsm) and source.
- The percentage mentioned in Table 1 (Page 5 and Line 220) is the percentage value by wt/wt, vol/vol, or else? This needs to be defined to ensure the repeatability of the coating formulation.
- The authors should discuss Figure 4 more in their writings.
- The scale bars are missing in each of the Figure 4 and Figure 7 panels.
- Lastly, the conclusions are too large, and it should rewire and mention the key findings for readers' easy understanding.
Author Response
Responses to reviewer 1.
The authors reported new alternatives in the valorisation of Eucalyptus globulus by-products for the textile industry. This study focuses how BLE particles can be incorporating in textile coating treatment which is of interest for the sustainable textile production. I recommend the publication of this work in Polymers after very minor revisions.
Some questions/comments are listed below:
Comment 1 The fabric composition is missing in the Raw Material section (Page 3 and Line 133). The authors should also add the fabric's weight (gsm) and source.
Thank you for your comment. The fabric composition, origin and weight have been added as suggested.
Comment 2. The percentage mentioned in Table 1 (Page 5 and Line 220) is the percentage value by wt/wt, vol/vol, or else? This needs to be defined to ensure the repeatability of the coating formulation.
Thanks for the suggestions. The percentage is wt/wt and was changed in Table 1.
Comment 3. The authors should discuss Figure 4 more in their writings.
Thank you for your suggestions. The following text was added for clarification:
“As for the non-fibrous particles, the stomatal pores of the eucalyptus leaves could be distinguished in the higher granulometry samples (BLE200 and BLE500) [21], but not in the lower granulometry samples which is due to the higher degree of grinding.”
Comment 4. The scale bars are missing in each of the Figure 4 and Figure 7 panels.
Thanks to suggestions, scale bars have been added to all SEM images in figures 4 and 7.
Comment 5. Lastly, the conclusions are too large, and it should rewire and mention the key findings for readers' easy understanding.
Thank you for your suggestion, the text of the conclusion has been modified as follows:
“The study demonstrates the potential of using unseparated small branches and leaves as a formulation component in textile coatings. Replacing 10% of the polyurethane with forest by-products resulted in a new product with differential colour, feel and mechanical performance. The particles improved the tensile strength without losing elongation of the coated fabrics, making them suitable for high-performance footwear. The thermal properties of the coated fabrics were also evaluated, and it was shown that the addition of BLE particles increased the residual mass at 600°C by more than four times. The transfer of the antioxidant properties of the eucalyptus particles to the textile product was confirmed, providing a functional textile product with potential applications in the medical, pharmaceutical and sports sectors. Further research could explore the economic and environmental benefits of using forest by-products in textile manufacturing. Overall, the study highlights the potential of using forest by-products in the development of new functional textile products”
Reviewer 2 Report
Comments and Suggestions for Authors
The manuscript entitled "New alternatives in the valorisation of Eucalyptus globulus by-2 products for the textile industry" is interesting and has got the scientific merit for publication after some major revisions are made.
- Remove the full stop in the title.
- Abstract needs to be more concise and should reflect the research gap, key findings of the research especially quantitative data.
- It is good if the authors can use some SDG related keywords.
- BLE is a non-standard abbreviation, if it is used as a code to clearly state it in the brackets.
- Line 60: The sentence like our research team is not scientifically appropriate 'mention as as reported' in the previous studies and cite the paper.
- There is a total lack of connectivity between each paragraph of the introduction. It should be clearly modified to incorporate the research gap.
- Section 2.4 : The brackets can be avoided as already it is mentioned in the following texts.
- 2.4.4. Extraction Yield should be changed as ‘Extraction Yield’
- DPPH assay: Change the % antioxidant efficiency as ‘DPPH radical scavenging activity (%)’
- Give the full form of FRAP initially
- What is the standard used for antioxidant tests? It should be plotted in the graph to give a comparison.
- The results of the characterization part should be properly discussed with minimum 3 published literature. It is lacking in several places.
- Conclusion: Present as a single paragraph with the key findings and future prospects
- There are several sentence errors in the manuscript that need to be corrected.
Author Response
Responses to reviewer 2.
The manuscript entitled "New alternatives in the valorisation of Eucalyptus globulus by-products for the textile industry" is interesting and has got the scientific merit for publication after some major revisions are made.
Comment 1
Remove the full stop in the title.
Thank you for your comment. The full stop has been removed.
Comment 2
Abstract needs to be more concise and should reflect the research gap, key findings of the research especially quantitative data.
Thank you for your comments. The abstract was revised, and quantitative data was added.
Comment 3
It is good if the authors can use some SDG related keywords.
Thanks for the suggestions, some key words related to the SDGs have been added.
Comment 4
BLE is a non-standard abbreviation, if it is used as a code to clearly state it in the brackets.
Thanks for the suggestions, BLE was placed in brackets throughout the document.
Comment 5
Line 60: The sentence like our research team is not scientifically appropriate 'mention as as reported' in the previous studies and cite the paper.
Thank you for your comment, the text was changed as the reviewer suggested.
Comment 6
There is a total lack of connectivity between each paragraph of the introduction. It should be clearly modified to incorporate the research gap.
Thank you for your suggestion, the introduction was changed to highlight the research gap
Comment 7
Section 2.4 : The brackets can be avoided as already it is mentioned in the following texts.
Thank you for your comment, the text was changed as the reviewer suggested.
Comment 8
2.4.4. Extraction Yield should be changed as ‘Extraction Yield’
Thank you for your comment, the text was changed as the reviewer suggested. Comment 9
DPPH assay: Change the % antioxidant efficiency as ‘DPPH radical scavenging activity (%)’
Thank you for your comment, the text was changed as suggested.
Comment 10
Give the full form of FRAP initially
Thank you for your comment, the text was changed as suggested.
Comment 11
What is the standard used for antioxidant tests? It should be plotted in the graph to give a comparison.
Thank you for your comment, The aim of the work is to evaluate the antioxidant capacity of fabrics coated with BLE particles, and the standards used are the textile substrate (cotton knit) and the fabric coated with the PUR formulation without BLE particles (PB).
Comment 12
The results of the characterization part should be properly discussed with minimum 3 published literature. It is lacking in several places.
Thank you for your comment, additional references have been added in the characterisation section.
Comment 13
Conclusion: Present as a single paragraph with the key findings and future prospects
Thank you for your suggestion, the text of the conclusion has been modified as follows:
“The study demonstrates the potential for using unseparated small branches and leaves as a component of textile coatings. Replacing 10% of the polyurethane with forest by-products resulted in a new product with differential colour, feel and mechanical performance. The particles improved the tensile strength of the coated fabrics without reducing their elongation, making them suitable for high-performance footwear. The thermal properties of the coated fabrics were also evaluated, and it was shown that the addition of BLE particles increased the residual mass at 600°C by more than four times. The transfer of the antioxidant properties of the eucalyptus particles to the textile product was confirmed, providing a functional textile product with potential applications in the medical, pharmaceutical and sports sectors. Further research could explore the economic and environmental benefits of using forest by-products in textile manufacturing. Overall, the study highlights the potential of using forest by-products in the development of new functional textile products.”
Comment 14
There are several sentence errors in the manuscript that need to be corrected.
Thank you for your suggestion. The document has been revised and the sentence errors detected have been corrected.
Round 2
Reviewer 2 Report
Comments and Suggestions for Authors
All the suggested modifications are made by author and team and the manuscript may be accepted .